# Ghrelin-induced neuronal NPY promotes brain metastasis in lung cancer patients with low BMI

Abhishek Tyagi [1], Shih-Ying Wu[1], Jee-Won Kim[1], Ravindra Pramod Deshpande [1], Kerui Wu [2], Eleanor C. Smith[1], Giuseppe L. Banna [3] & Kounosuke Watabe [1] ✉

Obesity is a known risk factor for many cancers, yet recent studies reveal a paradoxical association between low body mass index (BMI) and increased brain metastasis in lung cancer—referred to as the "obesity paradox," with unclear molecular mechanism(s). Here, we show a significantly higher incidence of brain metastasis in low-BMI lung cancer patients compared to those with high-BMI or other cancer brain metastasis in a pan-analysis of 7628 patients. Mechanistically, low BMI activates ghrelin-GHSR signaling, increasing neuronal neuropeptide Y (NPY) secretion, which promotes tumor metabolic reprogramming via NPY-Y5R, facilitating brain colonization. Elevated plasma ghrelin levels in cancer-free low-BMI subjects suggest its potential as a prognostic biomarker for predicting brain metastasis. Notably, targeting NPY-Y5R or reversing low BMI effectively suppresses brain metastasis, supporting its pro-metastatic role. These findings provide a strong rationale for developing targeted interventions to treat or prevent brain metastasis in lung cancer patients with low BMI.

Brain metastasis is the most common intracranial tumor, predominantly arising from lung (40–50%), breast (15–25%), and melanoma cancers (5–20%), with an annual incidence of 70,000 to 400,000 cases in the US[1,2]. Despite advancements in molecularly targeted therapies, immunotherapies, stereotactic radiosurgery, or traditional surgery[3], the median survival ranges from 5 to 10 months[4], underscoring the need to better understand this disease as a means to develop effective therapies. Importantly, lung cancer patients with brain metastasis have poor prognosis, high mortality rate, and frequent incidence of tumor recurrence due to associated clinical risk factors, including younger age, female gender, smoking, adenocarcinoma histology, karnofsky performance status (KPS), and the location and number of brain metastasis[5–8]. Notably, recent clinical epidemiological studies have revealed a positive association between low body mass index (BMI) and a two-fold increased risk of brain metastasis in lung cancer patients, even after adjusting for other predictive risk factors, leading to lower survival rates compared to patients with brain metastasis from other primary tumors[9–13]. How low-BMI contributes to an increased risk of lung cancer metastasis, preferentially in the brain compared to other organs, while simultaneously conferring a protective effect on other primary tumors, remains largely understudied. This paradoxical inverse relationship, referred to as the "obesity paradox," necessitates further investigation to unravel the underlying mechanisms and nature of this phenomenon, particularly in the context of brain metastasis, which remains poorly understood. In this work, we address this question using a systematic approach, by analyzing the effects of low BMI-associated lung-to-brain metastasis. We show that low BMI induces neuronal NPY secretion and promotes brain metastasis in lung cancer by reprogramming cancer cell energy metabolism, thereby facilitating tumor progression.

[1]Department of Cancer Biology, Wake Forest University School of Medicine, Winston-Salem, NC 27157, USA. [2]Nanoscience Department, Joint School of Nanoscience & Nanoengineering, Greensboro, NC 27401, USA. [3]Portsmouth Hospitals University NHS Trust, Faculty of Science and Health, School of Pharmacy and Biomedical Sciences, University of Portsmouth, Portsmouth, UK. ✉e-mail: kwatabe@wakehealth.edu

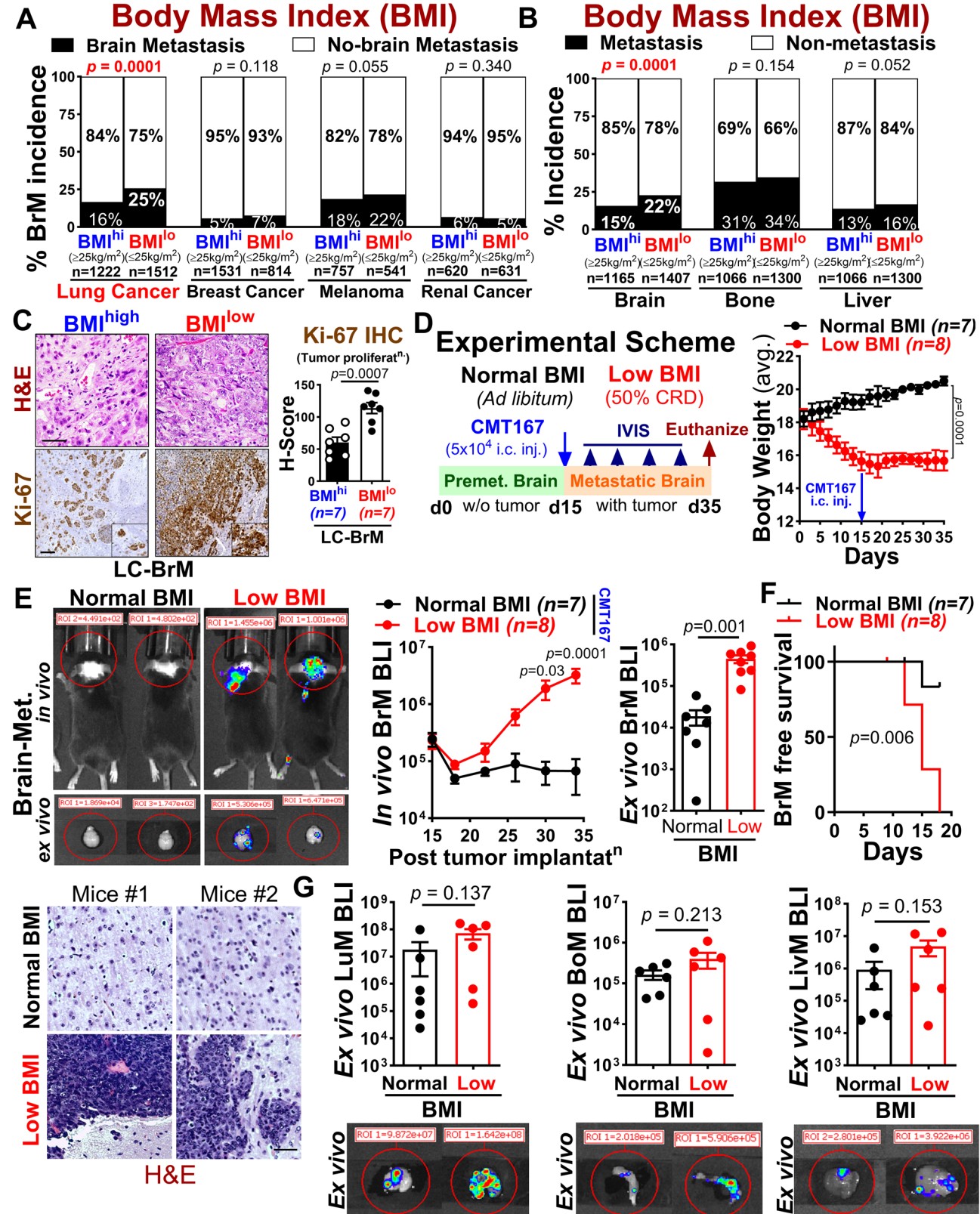

## Results

### Low body mass index (BMI) promotes brain metastasis in lung cancer

To understand the mechanism by which low BMI promotes brain metastasis preferentially in lung cancer compared to other cancer types, we performed a retrospective pan-analysis of multiple patient cohorts with brain metastasis from lung, breast, melanoma and renal

cancers with their BMI status, comprising a total of 7628 cancer patients[14–24]. We found a significant association between low BMI and a higher incidence of brain metastasis in lung cancer, in contrast to breast, melanoma and renal cancer (Fig. 1A). Notably, within a clinical cohort of metastatic lung cancer patients ($n = 2366$)[14–16,24], low BMI exhibited a significant association with brain metastasis compared to extracranial metastatic sites including bone and liver (Fig. 1B), and is

**Fig. 1 | Low body mass index (BMI) is associated with increased brain metastasis in lung cancer. A** Percentage of brain metastatic incidence across lung, breast, melanoma, and renal cancers ($n = 7628$), stratified by BMI status. Patients with BMI < 25 were categorized as low BMI (BMI[low]) and those with BMI > 25 as high BMI (BMI[high]) (two-sided Fischer's exact test). **B** Percentage of metastatic incidence in the brain, bone and liver within a lung cancer cohort ($n = 2366$) comprising both low and high BMI patients (two-sided Fischer exact test). **C** Left: Representative hematoxylin and eosin (H&E) and Ki-67 immunohistochemistry images of brain metastatic tumor lesions from lung cancer patients, stratified by BMI [Scale bar: 100μm (H&E), 50μm (Ki-67); insert: 2 x Zoom]. Right: Dot plot showing quantification of Ki-67+ cells in brain metastatic tumor lesions from low and high BMI patients using H-score ($n = 7$ each, unpaired two-tailed t-test). **D** Left: Schematic of in vivo metastasis assay. Immune-competent C57BL/6 (female, 5–6 week) mice were fed with a standard or 50% calorie-restricted diet (CRD; 12-hr/dark phase) for 15 days to establish low BMI mice, followed by intracardiac injection (i.c.) of murine lung cancer cell line, CMT167 cells ($5 \times 10^4$). Right: Average body weight curve during the experiment (two-way ANOVA with Tukey's test). **E** Top left: Representative in vivo (top) and ex vivo (bottom) BLI images of brain metastasis at endpoint. Middle/Right: In vivo and ex vivo BLI quantification of brain metastasis in normal ($n = 7$) and low BMI ($n = 8$) mice at endpoint (two-way ANOVA with Tukey's test (in vivo); unpaired two-tailed t-test (ex vivo). Bottom: Representative H&E images of brain metastatic tumor lesions from normal and low BMI mice (Scale bar: 50 μm). **F** Kaplan–Meier analysis of brain metastasis-free survival in normal ($n = 7$) and low BMI ($n = 8$) mice [log-rank (Mantel–Cox test)]. **G** Representative ex vivo BLI images and quantification of lung, liver and bone metastasis in normal and low BMI mice ($n = 6$/group) at endpoint (unpaired two-tailed t-test). All data are mean ± S.E.M. Source data are provided as a Source Data file.

also significantly associated with poor brain-metastasis free survival in a clinical cohort of metastatic lung cancer patients ($n = 513$)[25] (Supplementary Fig. 1A). Importantly, patients with lung cancer brain metastasis showed no significant difference with other clinical factors, including gender, race, smoking, *EGFR* mutation except for age (Supplementary Fig. 1B)[26–29]. To further gain insight into the underlying mechanism of low BMI-induced brain metastasis, we utilized clinical samples of brain metastatic lung cancer patients with BMI status and assessed Ki-67 immunoreactivity by immunohistochemistry. We found that brain metastatic patients with low BMI exhibited a higher number of Ki-67 positive tumor cells compared to those with high BMI, indicating an increase in tumor growth in brain metastatic lung cancer patients with low BMI (Fig. 1C). This contrasts with previous studies linking high BMI (obesity) with more aggressive tumor growth and increased cancer mortality in other cancer types[11,12,30,31]. To investigate how BMI impacts brain metastasis in vivo, we established a low-BMI mouse model through a regimen of 50% calorie restriction combined with time-restricted feeding to mimic human low BMI conditions (Supplementary Fig. 1C–E, and Supplementary Table 1) and examined its role in experimental metastasis by implanting CMT167 and LL2 mouse lung cancer cells in syngeneic C57BL/6 mice via intracardiac injection (i.c.) (Fig. 1D) followed by monitoring the metastatic burden by bioluminescence imaging (BLI) (Fig. 1E). We found that low BMI significantly increased the brain metastatic burden by more than 10-fold and exhibited significantly reduced brain-metastasis free survival compared to normal BMI (Fig. 1E–F, and Supplementary Fig. 1F–H). Importantly, low BMI has no significant impact on extracranial metastasis in comparison to normal BMI (Fig. 1G), consistent with our clinical observation. Taken together, these results strongly imply that low BMI plays a critical role in creating a favorable brain metastatic niche in lung cancer metastasis.

## Low BMI-mediated ghrelin activates neurons that promote tumor growth via NPY

What factors drive increased brain metastasis in patients with low BMI is an intriguing question. Prior studies have shown that plasma ghrelin, an orexigenic peptide regulating growth hormone secretion, food intake, and body weight through the growth hormone secretagogue receptor (*GHSR*)[32–39], is elevated in both lean and healthy individuals undergoing fasting or calorie restriction but decreased in obese individuals, suggesting a compensatory role in response to low BMI[40–44]. We also analyzed cancer-free subjects with known BMI data[43,45,46] and found a significant inverse correlation between high plasma ghrelin levels and low BMI compared to those with high BMI (Fig. 2A). Therefore, to understand the functional relationship between ghrelin and its potential impact on low BMI-induced brain metastasis, we treated normal BMI mice with ghrelin or administered a pharmacological GHSR inhibitor to low BMI mice. Experimental metastasis was assessed by intracardiac injection (i.c.) of CMT167 or LL2 mouse lung cancer cells into syngeneic C57BL/6 mice (Fig. 2B, and Supplementary

Fig. 2A–B). We found that ghrelin treatment significantly increased brain metastatic burden by over 10-fold, whereas GHSR inhibition reduced brain metastasis by more than 20-fold. These treatments also significantly affected survival, with ghrelin decreasing and GHSR inhibition increasing brain metastasis-free survival compared to the controls (Supplementary Fig. 2A–C). Notably, neither treatment impacted extracranial metastasis compared to controls (Supplementary Fig. 2A–D). Next, we examined the expression of the ghrelin receptor in various brain cell types as well as in parental and brain-tropic lung cancer cells. An enriched overexpression of *GHSR* receptor was found in neurons, consistent with a previous study[34], in contrast to other brain cell types or lung cancer cells, which exhibited low to negligible levels of *GHSR* expression (Fig. 2C, and Supplementary Fig. 3A). Furthermore, treating lung cancer cells with a physiological concentration of recombinant ghrelin (rGhrelin; 0.1nM-0.25 nM) showed no discernible effect on cancer cell proliferation and growth in a dose-dependent manner (Supplementary Fig. 3B). Conversely, conditioned media (CM) derived from primary neurons significantly increased cancer cell growth compared to CM from other brain cell types, such as astrocytes and microglia (Fig. 2D). To understand how neurons impact tumor cell proliferation and growth under low BMI conditions, we examined the effect of CM from primary neurons pre-exposed to physiological concentration of recombinant ghrelin (0.1nM-0.25 nM) on tumor cells. We found that CM from primary neurons pretreated with increasing doses of recombinant ghrelin protein significantly enhanced tumor cell proliferation and growth (Fig. 2E, and Supplementary Fig. 3C). To gain deeper insights into how low-BMI influences the brain secretome in response to elevated plasma ghrelin, we analyzed publicly accessible RNA-seq data from the normal human brain with BMI status ($n = 382$) from the GTEx (Genotype-Tissue Expression; dbGaP accession number phs000424.v8), in conjunction with the Human Protein Atlas (HPA) database[47], and found 7 low BMI-specific secretory genes in the brains of cancer-free subjects with low BMI (Fig. 2F, and Supplementary Table 2; unpaired two-tailed t-test). Further validation in human primary neurons pretreated with recombinant ghrelin showed a significantly enriched expression of neuropeptide Y (*NPY*) compared to other genes (Fig. 2G), and the expression was further increased with escalating doses of recombinant ghrelin (0.1nM-0.25 nM), primarily within neurons as opposed to other brain cell types (Supplementary Fig. 3D–E). Importantly, similar results were observed in tumor-free low BMI mice (from Fig. 1; panel D) as well as in normal brain tissue and brain metastatic tissue from H2030BrM xenografts, in contrast to normal BMI mice, non-brain tissue or primary lung cancer (Fig. 2H, Supplementary Fig. 3F–G). Additionally, conditioned media (CM) from ghrelin-treated primary neurons showed significantly higher NPY levels compared to controls or cancer cells (Fig. 2I, and Supplementary Fig. 3H). Notably, when lung cancer cells were treated with recombinant NPY (rNPY), they showed a dose-dependent increase in cell proliferation, growth kinetics and cell cycle progression. These results suggest that the tumor-promoting effect of

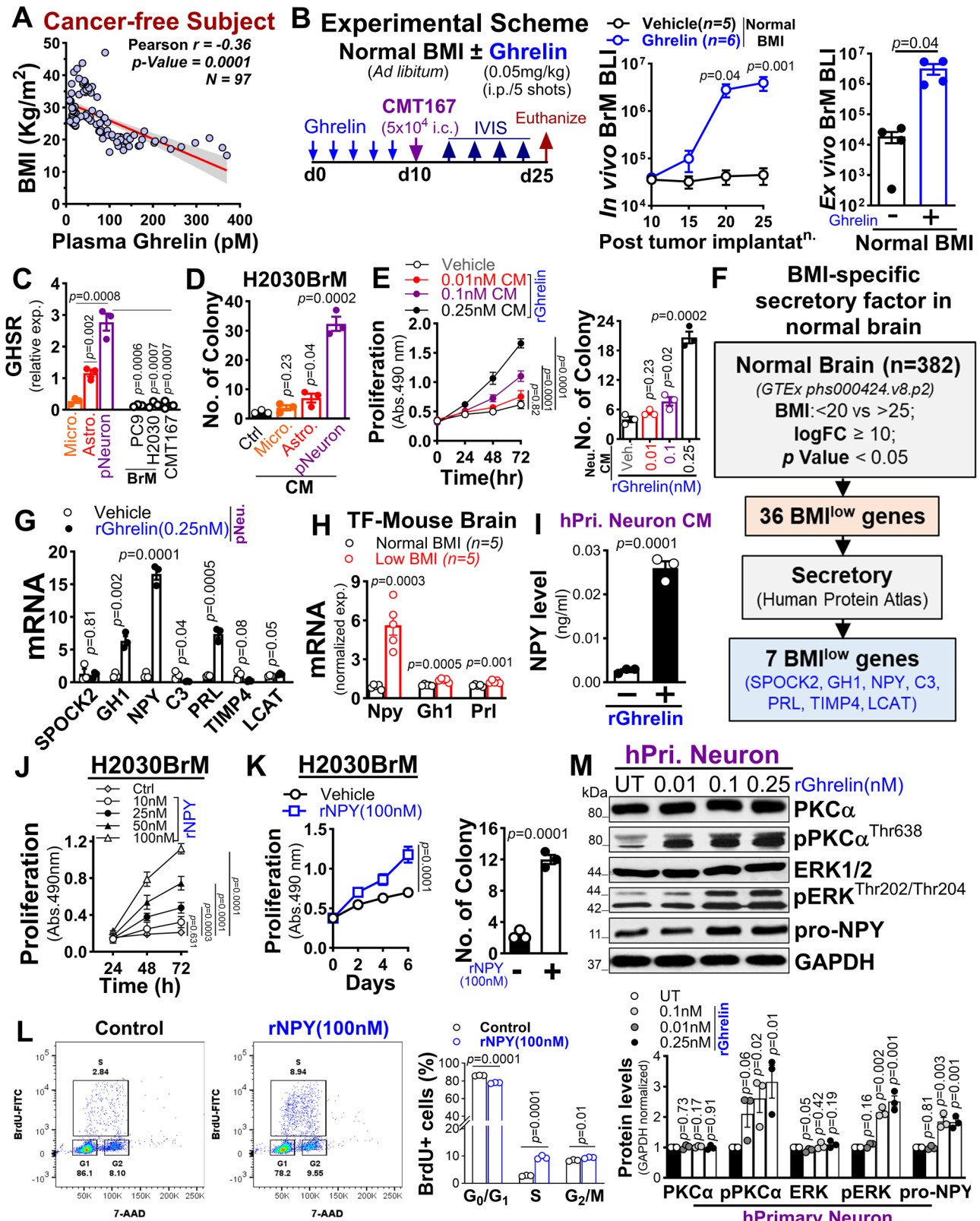

low BMI is likely mediated by ghrelin-activated neurons, which secrete NPY with a potential growth-promoting role in tumor cells (Fig. 2J–L, and Supplementary Fig. 3I). Next, to elucidate the molecular pathway responsible for ghrelin-induced NPY secretion, we focused on the PKCα/ERK pathway, previously implicated in neuronal NPY regulation in a ghrelin-dependent manner[48,49]. We observed a ghrelin-specific, dose-dependent increase in NPY protein levels, accompanied by an

elevated expression of its upstream effector protein (PKCα/pERK) in primary neurons (Fig. 2M). Notably, this expression was abrogated in primary neurons by a lower dose of a pharmacological ERK1/2 inhibitor (SCH772984), in contrast to cancer cells, which exhibited cell toxicity and growth suppression at higher doses (Supplementary Fig. 3J–L), providing further confirmation of the PKCα/ERK axis as the key signaling pathway responsible for ghrelin-mediated neuronal NPY

**Fig. 2 | Low BMI promote tumor growth via ghrelin activated neuronal NPY.**
**A** Pearson correlation between BMI and plasma ghrelin levels in cancer-free subjects ($n = 97$) using a publicly available dataset (two-tailed; 95% CI shaded). **B** Left: Schematic of in vivo metastasis assay. Female C57BL/6 mice (5-6 weeks) were fed with a standard diet and administered ghrelin (0.05 mg/kg; i.p.; 5 shots) for 10 days before CMT167 cells injection ($5 \times 10^4$; i.c.). In vivo (middle) and ex vivo (right) BLI quantification of brain metastasis in control ($n = 5$) and ghrelin-treated normal BMI mice ($n = 6$) [two-way ANOVA with Tukey's test (in vivo); unpaired two-tailed t-test (ex vivo)]. **C** Relative *GHSR* expression in human brain cells (HMC3, UC1, primary neurons) and lung cancer cell lines (PC9BrM, H2030BrM, CMT167) by qRT-PCR, normalized to *β-Actin* ($n = 3$ independent experiments, unpaired two-tailed t-test). **D** Colony formation of H2030BrM cells treated with or without CM from indicated brain cells ($n = 3$ independent experiments, unpaired two-tailed t-test). **E** MTS (left) and colony (right) assays of H2030BrM cells treated with or without CM from ghrelin-treated primary neurons (0.01 nM-0.25 nM; $n = 3$ independent experiments, two-way ANOVA with Tukey's test). **F** Strategy to identify low BMI-specific brain secretory factors using GTEx and HPA database ($n = 382$). **G** qRT-PCR of indicated genes in primary neurons treated with or without rGhrelin (0.25 nM, 12 h, normalized to *β-Actin*; $n = 3$ independent experiments, unpaired two-tailed t-test). **H** Expression of *Npy*, *Gh1* and *Prl* in brains of normal and low-BMI mice (Fig. 1D) by qRT-PCR (normalized to *β-Actin*; $n = 3$ independent experiments, unpaired two-tailed t-test). **I** NPY protein levels in CM from untreated and ghrelin-treated (0.25 nM) neurons by ELISA ($n = 3$ independent experiments, unpaired two-tailed t-test). **J, K** MTS and colony assays of H2030BrM cells treated with rNPY [10–100 nM, short-term (J); 100 nM, long-term (K)]; (J: $n = 3$ independent experiments, two-way ANOVA with Tukey's test; K: $n = 3$ independent experiments, unpaired two-tailed t-test). **L** Dot plot (left) and quantification (right) of BrdU-based cell cycle analysis of H2030BrM cells treated with or without rNPY (100 nM; $n = 3$ independent experiments, unpaired two-tailed t-test). **M** Western blot (upper) and ImageJ quantification (lower) of GHSR signaling proteins in neurons treated with or without rGhrelin (0.01-0.25 nM, 24 h; normalized to GAPDH; $n = 3$ independent experiments, unpaired two-tailed t-test). All data are mean ± S.E.M. Source data are provided as a Source Data file.

secretion via *GHSR*. Collectively, these results indicate the pro-tumorigenic role of neuronal NPY in promoting brain metastasis under low-BMI conditions, distinct from its physiological role in regulating feeding behavior and maintaining energy homeostasis.

## Neuronal NPY promotes brain metastasis via NPY5R (Y5R) signaling

To gain further insight into the functional role of low BMI-induced NPY in brain metastasis, we queried clinical GEO cohort data of lung[50], breast[51], and melanoma[52] cancers, comprising matched primary and brain metastasis samples to assess NPY receptor expression patterns. We found that brain metastatic lesions of lung cancer exhibited significantly higher expression of the *NPY5R* (*Y5R*) compared to other NPY receptors, and the expression of *Y5R* was selectively enriched in metastatic lung cancer, in contrast to metastatic breast and melanoma cancer (Fig. 3A). Additionally, increased expression of Y5R is associated with poor overall survival in lung cancer patients, attributed to its elevated levels in metastatic lung tumors compared to normal and primary lung tumors. In contrast, in breast cancer, *Y5R* expression is correlated with better overall survival, with higher expression observed in normal and primary tumors compared to metastatic breast tumors (Supplementary Fig. 4A). To further validate this observation, we analyzed multiple GEO cohort datasets, including clinical and preclinical samples of lung, breast, and melanoma metastatic tumors, as well as primary tumor xenografts (GSE123904 /GSE48433 /GSE14020/GSE50493)[53–56]. We found that *Y5R* was selectively expressed in metastatic tumors, primarily in the brain, in lung cancer, in contrast to metastatic tumors from breast and melanoma cancers (Supplementary Fig. 4B). As the Y5R has recently been reported to play a critical role in tumor progression[57,58], we next examined its expression in brain cells and lung cancer cells. We found that cancer cells exhibited a higher level of *Y5R* expression compared to brain cells, except neurons that showed basal expression as reported previously[59,60] (Supplementary Fig. 4C). We then examined clinical brain metastatic samples from lung cancer patients with BMI history. We observed significantly higher *Y5R* expression in lung cancer brain metastatic patients with low BMI compared to those with high BMI (Fig. 3B). Similar results were obtained in murine and human lung cancer cell lines, both at transcript and protein levels, and its expression was significantly upregulated in the cancer cells in the presence of recombinant NPY (Fig. 3C, and Supplementary Fig. 4D). Additionally, we analyzed RNA-seq data and performed Gene Set Enrichment Analysis (GSEA)[61] to evaluate neuropeptide receptor activity in matched primary and brain metastatic tissues obtained from a clinical study of lung cancer patients[50]. We found significantly elevated neuropeptide receptor signaling in brain metastatic tumors compared to primary tumors (Fig. 3D), suggesting that neuropeptide receptor activity may

play a crucial role in governing tumor cell aggressiveness and metastatic progression. NPY peptides and their receptors are known to influence cancer progression through distinct signaling pathways[62], but the underlying molecular mechanisms remain elusive. Therefore, we analyzed existing RNA-sequencing data of brain metastatic tumors from a lung cancer clinical cohort[50] and performed GSEA for oncogenic pathways by stratifying brain metastatic patients based on their *NPY* receptor expression. We found that brain metastatic patients with a high *NPY* receptor level (*Y5R*<sup>high</sup>) showed activated MAPK signaling signatures, while patients with a low NPY receptor level (*Y5R*<sup>low</sup>) displayed signatures associated with the Notch pathway (Fig. 3E). This indicates potential divergent roles of Y5R signaling in brain metastasis. In support of this finding, we noted a positive correlation between *Y5R* and *ERK5* expressions compared to *p38* and *JNK*, members of the MAPK signaling pathway in brain metastatic patient cohort (Fig. 3F). To further characterize the role of the Y5R in promoting low BMI-induced neuronal NPY-mediated tumor growth, we utilized CRISPR/Cas9 technology to knockout *Y5r* in a murine CMT167 lung cancer model. Two independent clones with a stable *Y5r* knockout (Y5rKO) were established, as described in the "Methods" section. The knockout of *Y5r* did not affect cancer cell growth upon recombinant NPY treatment, as compared to scrambled knockout (ScrKO) control cells (Supplementary Fig. 4E–G). We then performed an in vivo experimental metastasis by intracardiac injection of luciferase-expressing ScrKO or Y5rKO cancer cells into syngeneic C57BL/6 mice with normal or low BMI conditions. We found that loss of the Y5r significantly decreased the brain metastatic tumor growth and resulted in a significant increase in brain-metastasis free survival in low BMI mice compared to the control mice implanted with scrambled knockout cancer cells (Fig. 3G–I). The *Y5r* knockout did not show a significant impact on extracranial metastasis under low BMI conditions as compared to the control mice with normal BMI (Fig. 3J). These results are consistent with our clinical observations, highlighting the significant impact of low BMI on brain metastasis. Overall, these results suggest that enhanced neuronal activity under low BMI conditions promotes cancer cells proliferation and growth through the NPY/Y5R signaling axis in the brain during the early stages of metastatic colonization.

## Neuronal NPY promotes metabolic reprogramming in tumor cells by upregulating lipogenesis via Y5R/ERK5/SREBP2/FASN pathway

Emerging studies on neuropeptide receptor signaling have highlighted its physiological role in the central regulation of metabolic homeostasis[63]. In addition, recent evidence also indicates that differences in nutrient availability in brain tissue relative to other tissue can necessitate metabolic adaptation or competition by cancer cells for growth in the brain[64,65]. However, the intricate mechanisms underlying

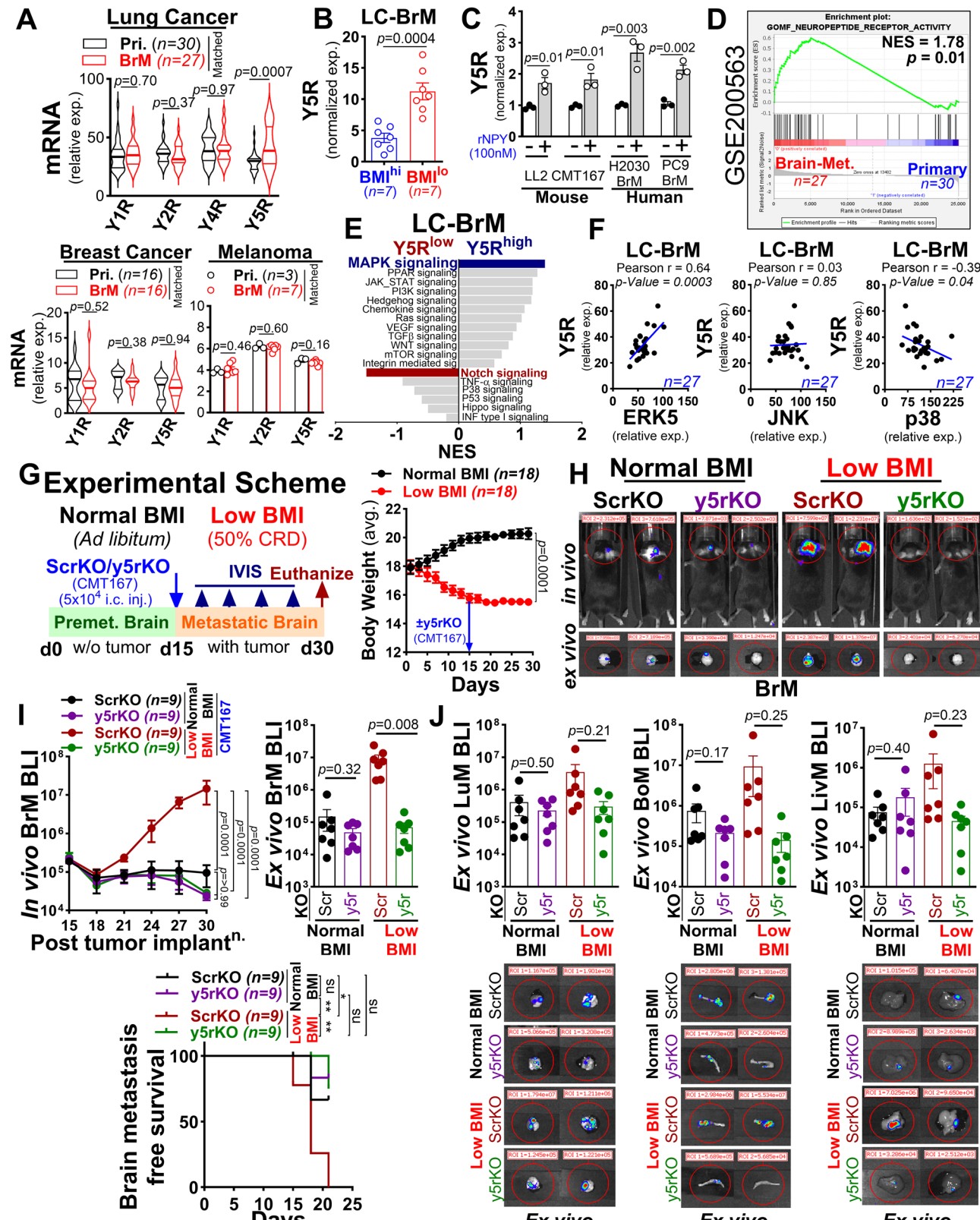

activated Y5R influence on tumor cell metabolism remain elusive. To address this question, we queried existing RNA-seq. data of matched primary and brain metastatic tissues from a clinical cohort of lung cancer[50] and evaluated major metabolic pathways by GSEA. We found that fatty acid metabolism was highly enriched in brain metastatic tumors, while glycolytic pathways were activated in primary tumors

(Fig. 4A). Similar results were obtained in another clinical cohort of patients with lung cancer brain metastasis (Supplementary Fig. 5A). Interestingly, brain metastatic patients with a high NPY receptor level (Y5R$^{high}$) showed activated fatty acid β-oxidation signatures, whereas patients with a low NPY receptor level (Y5R$^{low}$) showed glycolysis signatures similar to those found in primary tumors (Fig. 4A). These

**Fig. 3 | Low BMI-mediated neuronal NPY promotes brain metastasis via Y5R signaling. A** Violin and dot plots analysis showing NPY receptor expression in matched brain metastatic samples from lung, breast, and melanoma cancer patients using GEO datasets (GSE200563 - upper; GSE125989 and GSE44660 -lower panels; unpaired two-tailed t-test). **B** qRT-PCR analysis of *Y5R* expression in brain metastatic tissue from lung cancer patients with low and high BMI (*n* = 7/group), normalized to *β-Actin* (unpaired two-tailed t-test). **C** qRT-PCR analysis of *Y5R* expression in human and mouse lung cancer cell lines treated with 100 nM rNPY, normalized to *β-Actin* (*n* = 3 independent experiments, unpaired two-tailed t-test). **D** Gene Set Enrichment Analysis (GSEA) of neuropeptide receptor activity in matched lung (*n* = 30) and brain metastatic tumors (*n* = 27) from GSE200563, using pre-ranked MSigDB v5.1gene sets. Enrichment scores generated by random forest-based ranking; *p*-values indicate statistical significance. **E** Bar plot showing enriched signaling pathways in Y5R^high and Y5R^low lung cancer brain metastasis (GSE200563). **F** Pearson correlation between *Y5R* and *MAPK* family members in brain metastatic

patients (*n* = 27, GSE200563; two-tailed); *Y5R-ERK5* (*p* = 0.0003), *Y5R-JNK* (*p* = 0.04) and *Y5R-p38* (*p* = 0.853). **G** Left: Schematic of in vivo metastasis assay. Female C57BL/6 mice (5–6 weeks) were fed with a standard or 50% CRD (12-hr/dark phase) for 15 days before ScrKO and y5rKO CMT167 cells injection (5 × 10⁴; i.c.). Right: Average body weight curve (two-way ANOVA with Tukey's test). **H** Representative in vivo (top) and ex vivo (bottom) BLI images of brain metastasis at endpoint. **I** Top: In vivo (*n* = 9/group; left) and ex vivo (*n* = 7/group; right) BLI quantification of brain metastasis in ScrKO and y5rKO implanted mice [two-way ANOVA with Tukey's test (in vivo); unpaired two-tailed t-test (ex vivo)]. Bottom: Kaplan–Meier analysis of brain metastasis-free survival in ScrKO and y5rKO implanted mice [*n* = 9/group; log-rank (Mantel–Cox test)]. **J** Representative ex vivo BLI images and quantification of extracranial metastasis in ScrKO and y5rKO implanted mice (*n* = 7/group) at endpoint (unpaired two-tailed t-test). All data are mean ± S.E.M. Source data are provided as a Source Data file.

results suggest the possibility that low BMI-induced neuronal NPY promotes metabolic reprogramming of tumor cells via tumor cell-specific Y5R. Therefore, we measured fatty acid oxidation (FAO) driven oxygen consumption rate (OCR) and glycolytic rate (ECAR) in control (ScrKO) and Y5rKO cancer cells in the presence or absence of recombinant NPY treatment (100 nM). We found a significant upregulation of FAO-driven OCR relative to glycolysis in NPY-treated control cells compared to untreated or Y5rKO cancer cells (Fig. 4B). We then examined whether NPY-induced fatty acid metabolism also exhibits an increased accumulation of lipids in the tumor cells. Using Oil-red O, a lipophilic dye that labels neutral lipids, we found a marked increase in the staining of lipid droplets in NPY-treated control cells compared to untreated or Y5rKO cancer cells (Fig. 4C, and Supplementary Fig. 5B). This result suggests that NPY promotes de novo lipogenesis in cancer cells, thereby enhancing tumor cell proliferation and survival. We therefore evaluated the expression level of SREBP2, a lipogenic regulator[66] in the presence or absence of recombinant NPY protein in conjunction with Y5R^high-specific pathways (PPAR, JAK/STAT3, PI3K) that were found to be enriched in lung cancer brain metastatic clinical samples (Fig. 3, panel E). Interestingly, among all, only the active SREBP2 level was found to be upregulated in NPY-treated control cells, in contrast to untreated or Y5rKO cancer cells as well as in NPY-treated parental lung cancer cells (Fig. 4D-E). This upregulation is positively correlated with ERK5 along with SREBP2, FASN and CPT1 in brain metastatic tumor tissue (*n* = 19)[67] (Fig. 4F). Similar results were observed in TCGA lung cancer database (Supplementary Fig. 5C) and validated in control cells (ScrKO) treated with or without recombinant NPY protein (100 nM), in the presence or absence of the ERK5 inhibitor (100 nM), compared to untreated or Y5rKO cancer cells (Fig. 4G-H), providing further evidence for the regulatory role of NPY in modulating the ERK5/SREBP2 signaling pathway. Additionally, by querying an existing RNA-seq. dataset of lung cancer cells cultured with or without glucose (GSE199089)[68], we found differential expression of genes controlling the fatty acid and cholesterol synthesis pathways in glucose-starved lung cancer cells (mimicking a low BMI condition) compared to cells exposed to glucose alone (Supplementary Fig. 5D). Next, to elucidate the molecular mechanism underlying NPY-mediated Y5R upregulation, we explored potential regulatory factors involved in the Y5R gene. Our analysis identified a putative SREBP2 transcription factor binding site within the 5' region of Y5R (-515 relative to TSS), suggesting that SREBP2 may play a key role in mediating this effect. Furthermore, our clinical data from lung cancer brain metastatic patients showed a significant positive correlation between Y5R and ERK5 expression, as well as between ERK5 and SREBP2. Therefore, to validate these findings, we ectopically expressed either ERK5 or SREBP2 in cancer cells, with or without recombinant NPY, in the presence or absence of ERK5 or SREBP inhibitors. We found that overexpression of either ERK5 or SREBP2 led to significant upregulation of Y5R expression compared to controls, while their specific inhibitors

significantly reduced the expression of Y5R (Fig. 4I). To further validated these results, we examined cancer cell proliferation in the presence or absence of CPT1 inhibitor. We found that CPT1 inhibition significantly reduced proliferation compared to controls (Supplementary Fig. 5E). These findings, consistent with our Y5R knockout metabolomic data, support the notion that NPY-induced metabolic changes via the SREBP2/FASN/CPT1 axis contribute to the proliferative effects during the early stages of metastatic colonization in the brain.

## Reversing the low BMI phenotype or blocking Y5R suppresses brain metastasis in lung cancer

Because low BMI plays a significant role in the progression of brain metastasis in lung cancer, we examined whether reversing the low-BMI phenotype upon resuming a normal diet, either before or after cancer cell inoculation, would impact brain metastasis. We took a weight-loss and weight-regain diet intervention approach in a brain metastasis model by intracardially injecting CMT167 mouse lung cancer cells into syngeneic C57BL/6 mice to assess brain metastasis progression (Fig. 5A, and Supplementary Fig. 6A). Notably, we found that reversing the low BMI phenotype resulted in a significant increase in body weight in the low-BMI mice along with a significant decrease in brain metastatic burden by more than 20-fold compared to the control group that was maintained on a low-BMI condition (Fig. 5B, and Supplementary Fig. 6B). This led to significantly enhanced brain metastasis-free survival compared to the control group (Fig. 5C, and Supplementary Fig. 6B). Furthermore, mice exposed to the weight-regain intervention regimen did not show any significant decrease in extracranial metastatic sites (lung, bone, and liver) compared to brain metastasis (Fig. 5D, and Supplementary Fig. 6C), consistent with our clinical observation. To further validate our results, we examined the circulating levels of plasma ghrelin during the low BMI reversal at different time points throughout the course of brain metastasis progression. We found a significant decrease in plasma ghrelin levels in the mice that were subjected to weight-regain intervention on day 25 compared to the low-BMI control mice from day 15 to day 25. We also found a persistent decrease in the plasma ghrelin levels within the weight-regain mice post-cancer cell implantation (day 35), in contrast to low-BMI control mice, which showed a significant increase in plasma ghrelin levels during metastatic outgrowth (Fig. 5E), suggesting that low BMI is a severe but modifiable risk factor for brain metastasis in lung cancer. The aforementioned results indicate that ghrelin-mediated low-BMI activates neuronal NPY, which supports Y5R^high cancer cells in metastasis to the brain, providing direct evidence for the crucial role of the NPY/Y5R axis in brain metastasis. Therefore, identifying an interventional agent that specifically blocks NPY's effect on tumor cells would be a promising therapeutic approach to treat lung cancer brain metastasis. Accordingly, we focused on the FDA-approved drug, MK-0557, a potent, highly selective, orally active, blood-brain barrier (BBB) permeable Y5R antagonist that underwent a

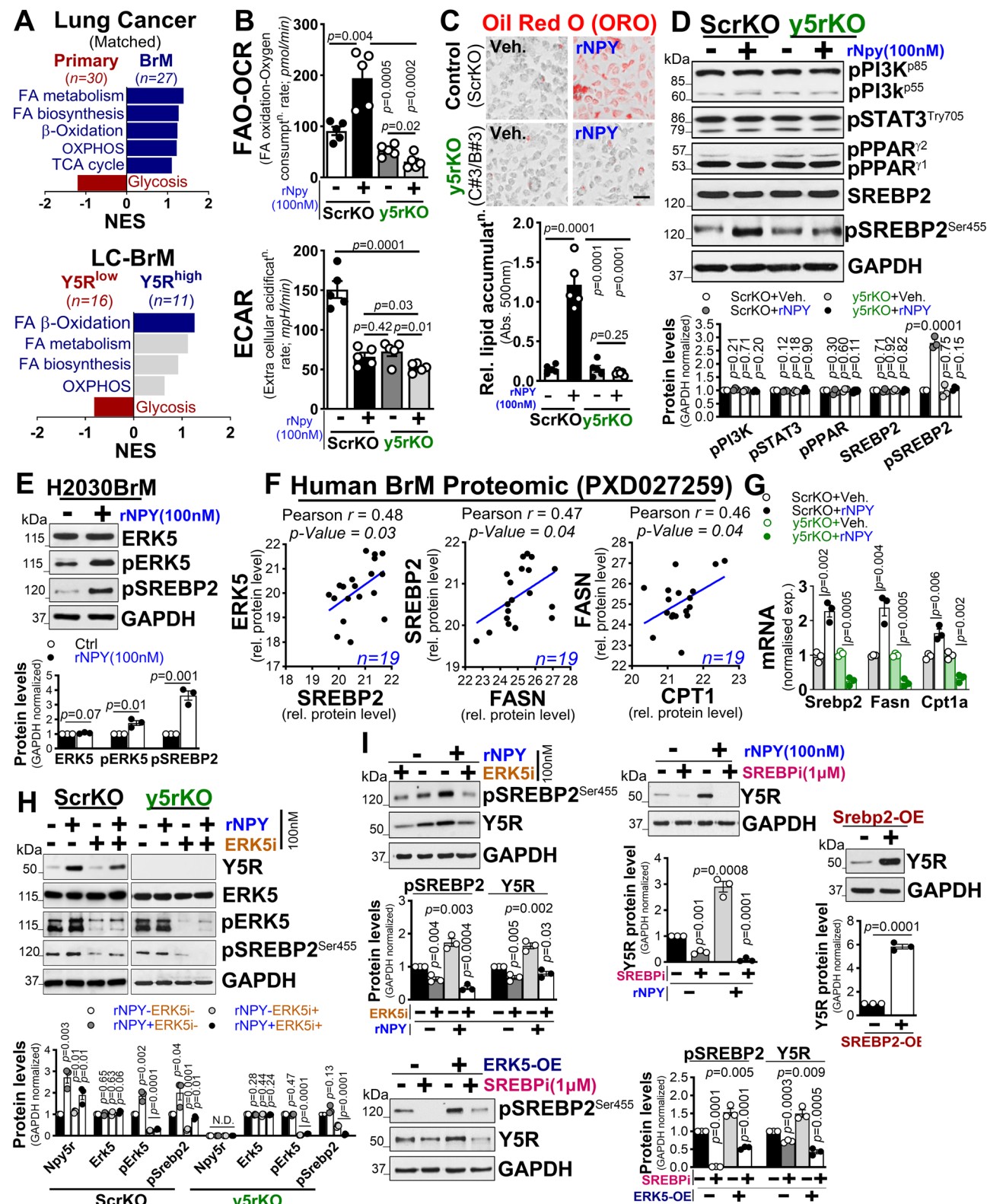

clinical trial aiming to treat obesity but did not induce clinically meaningful weight loss[69,70]. We, therefore, tested the in vivo efficacy of MK-0557 (Y5R antagonist) on brain metastasis by implanting CMT167 lung cancer cells into syngeneic C57BL/6 mice followed by oral administration of MK-0557 (1 mg/kg; 6 shots till endpoint) under low-BMI condition (Fig. 5F). We found that MK-0557 significantly decreased low-BMI induced brain metastasis and thereby increased brain metastasis-free survival without any change in body weight (Fig. 5G–I).

Furthermore, MK-0557 treatment did not induce any notable toxicity or weight loss in the treated mice (Supplementary Fig. 6D-E) and only showed effects on cancer cell viability and growth at higher doses compared to Y5R-KO cancer cells (Supplementary Fig. 6F). Importantly, ghrelin pre-treated neuron-conditioned media failed to rescue cancer cell proliferation under MK-0557 treatment suggesting inhibition of the NPY-specific growth-promoting effect on tumor cells (Supplementary Fig. 6G). We also examined the number of lipid

**Fig. 4 | Low BMI-induced neuronal NPY upregulates lipogenesis via the Y5R/SREBP2/FASN signaling axis to fuel tumor cell growth. A** Metabolic GSEA was performed in matched primary lung ($n = 30$) and brain metastatic ($n = 27$) tumors (top) and between Y5R^high and Y5R^low brain metastatic lesions (bottom, GSE200563). Gene sets were derived from Gene Ontology (GO:0006635; GO:0006099; GO:0061621, GO:0046949, GO:0006119, GO:0006631)[153]. **B** FAO-driven oxygen consumption (OCR) and glycolytic rate (ECAR) were measured in ScrKO and y5rKO CMT167 cells pretreated with or without rNPY (100 nM) ($n = 5$ independent experiments, unpaired two-tailed t-test). **C** Representative images (top) and quantification (bottom) of Oil Red O staining showing lipid accumulation in ScrKO and y5rKO lung cancer cells treated with or without rNPY (100 nM, 12 h; Scale bar: 50 μm; $n = 5$ independent experiments; unpaired two-tailed t-test). **D** Top: Western blot of SREBP2 and Y5R^high-enriched key oncogenic pathways in ScrKO and y5rKO CMT167 cells treated with or without rNPY (100 nM, normalized to GAPDH; $n = 3$ independent experiments, unpaired two-tailed t-test). Bottom: Band intensity quantification by ImageJ (unpaired two-tailed t-test). **E** Top: H2030BrM cancer cells treated with or without rNPY (100 nM) and immunoblotted for activated ERK5 and SREBP2, normalized to GAPDH ($n = 3$ independent experiments, unpaired two-

tailed t-test). Lower: Band intensity quantification by ImageJ (unpaired two-tailed t-test). **F** Pearson correlation between ERK5-SREBP2, $p = 0.036$; SREBP2-FASN, $p = 0.041$ and FASN-CPT1, $p = 0.047$ in lung cancer brain metastatic patients using a proteomic dataset (PXD027259; $n = 19$; two-tailed)[67]. **G** Expression of *Srebp2*, *Fasn* and *Cpt1a* in ScrKO and y5rKO CMT167 cells treated with or without rNPY (100 nM), by qRT-PCR, normalized to *β-Actin* n ($n = 3$ independent experiments, unpaired two-tailed t-test). **H** Top: Cells from panel G were further immunoblotted for Y5R, pERK5 and SREBP2 in the presence or absence of rNPY (100 nM) or ERK5 inhibitor (100 nM each), GAPDH normalized ($n = 3$ independent experiments, unpaired two-tailed t-test). Bottom: Band intensity quantification by ImageJ (unpaired two-tailed t-test). **I** Top: LL2 cells were transfected with or without ERK5/SREBP2 expression plasmids or treated with or without SREBP (Fatostatin HBr) or ERK5 inhibitor (ERK5-IN-1), in presence or absence of rNPY (100 nM) were immunoblotted for activated SREBP2 and Y5R, GAPDH normalized ($n = 3$ independent experiments, unpaired two-tailed t-test). Bottom: Band intensity quantification by ImageJ (unpaired two-tailed t-test). All data are mean ± S.E.M. Source data are provided as a Source Data file.

droplets in brain metastatic tumor lesions from control and MK-0557 treated low-BMI mice by LipidSpot staining. We found that MK-0557 treatment significantly reduced the number of Ki-67 positive tumor cells as well as lipid droplets in brain metastatic tumor lesions of treated mice compared to the control mice (Fig. 5J, Supplementary Fig. 6H). Overall, these results provide a rationale for advocating the maintenance of normal body weight and exploring the blockade of NPY/Y5R interactions (Supplementary Fig. 7A–B). This could help determine its strength as a promising treatment approach for potential clinical intervention in treating brain metastasis in lung cancer patients based on BMI status. In summary, we propose the underlying mechanism by which, under the influence of ghrelin, low BMI induces neuronal activation and secretion of NPY via GHSR/PKCα/ERK axis, which binds to Y5R on tumor cells, triggering a metabolic switch from glucose to fatty acids, thereby promoting brain metastatic progression (Fig. 5K).

## Discussion

A large body of epidemiological data has demonstrated a strong association between high BMI and an increased risk of cancer mortality across various cancers[71–74]. On the contrary, a negative correlation has been observed between high BMI and the progression or mortality of lung cancer[75–77]. Furthermore, meta-analyses conducted over the past decade using large cohort databases have identified the existence of the "obesity paradox" in advanced lung cancer compared to other cancers regarding the relationship between BMI and incidence[78–81]. However, the reason why high BMI is associated with a lower incidence of lung cancers and lower cancer-specific mortality in comparison to other cancers remains unclear. This raises the intriguing question of whether low BMI serves as a surrogate for cachexia, particularly in advanced lung cancer, which has a reported incidence of 40-50%, and is associated with poor overall survival[82]. BMI alterations are often attributed to an imbalance between energy intake and total energy expenditure (TEE)[83], which may increase energy metabolism. This association could stem from patients' failure to sufficiently increase their energy intake, leading to altered body composition that contributes to cachexia development. This trend is evident in a recent metastatic lung cancer clinical trial (NCTO314)[84] and among healthy underweight adults[85]. These observations suggest that cancer cachexia may not be a causative factor but rather a consequence of low body weight (anorexia) due to hypermetabolism in metastatic lung cancer as recently reported[86]. However, the paradoxical prognostic and predictive relationships between low and high BMI in lung cancer patients in the metastatic setting have not been well understood. Interestingly, recent clinico-epidemiological studies have shown a positive correlation between low BMI and a 2-fold increased risk of brain metastasis in lung cancer patients as compared to those with a high BMI

or other cancer types, which is further correlated with poor overall survival[9–13]. Nonetheless, the nuanced relationship between low BMI and brain metastasis warrants further research, particularly in understanding the underlying molecular mechanisms, with potential implications for the management and prognosis of metastatic lung cancer patients with low BMI. We demonstrated that low BMI serves as a robust predictor of adverse outcomes, suggesting a potential association between nutritional status, metabolic alterations, and reduced survival outcomes, with an unfavorable prognosis for metastatic lung cancer, which profoundly impacts the patient's quality of life. Overall, our findings offer potential implications for early identification and intervention in patients with low BMI. This underscores the necessity for tailored interventions, including BMI monitoring upon initial admission, effective nutritional support, and dietary modification to address metabolic dysregulation, which could lead to improved treatment outcomes and prognosis in patients with metastatic lung cancer and low BMI.

Ghrelin is an orexigenic peptide primarily secreted by the stomach, playing a crucial role in regulating appetite, body weight, energy expenditure, and growth hormone secretion[34,35,38,87]. This peptide modulates neuronal activity by traversing the blood-brain barrier (BBB) and blood-cerebrospinal fluid barrier (BCB)[33,37,39,88]. Moreover, ghrelin directly interacts with neurons via multiple mechanisms through its receptor, GHSR (a G-protein-coupled secretagogue receptor)[36], and promotes neurogenesis in diet-restricted mice[89,90]. Conversely, ghrelin exhibits an inhibitory effect on other brain cell types expressing GHSR, such as astroglia and microglia, to protect neurons from the adverse effects of pro-inflammatory factors released during neurodegenerative disease or injury processes[91–94]. In addition to its neuroprotective role, ghrelin also plays a role in synapse formation and the generation of long-term potentiation, contributing to enhanced spatial learning and memory[95], as well as additional physiological functions such as psychological stress, mood, anxiety[96], depression[97], and aging[98]. Interestingly, increased serum ghrelin levels have been previously reported to induce body weight loss (anorexia) but not muscle mass loss (cachexia) in lung cancer patients[99,100]. Additionally, prior studies have indicated elevated circulating ghrelin levels in lean and healthy individuals undergoing fasting, calorie restriction regimes, or exercise regimens, as opposed to individuals with normal weight, overweight, or non-exercising status. This reflects a negative feedback mechanism to maintain energy homeostasis in non-conducive environments[40,42–44,101–103]. Notably, we found a significant inverse correlation between high plasma ghrelin levels and low BMI in cancer-free subjects and in mice with brain metastasis, with its levels significantly decreased in response to a weight-loss and weight-regain diet intervention approach in vivo, leading to increased brain-metastasis free survival. These findings suggest that circulating ghrelin

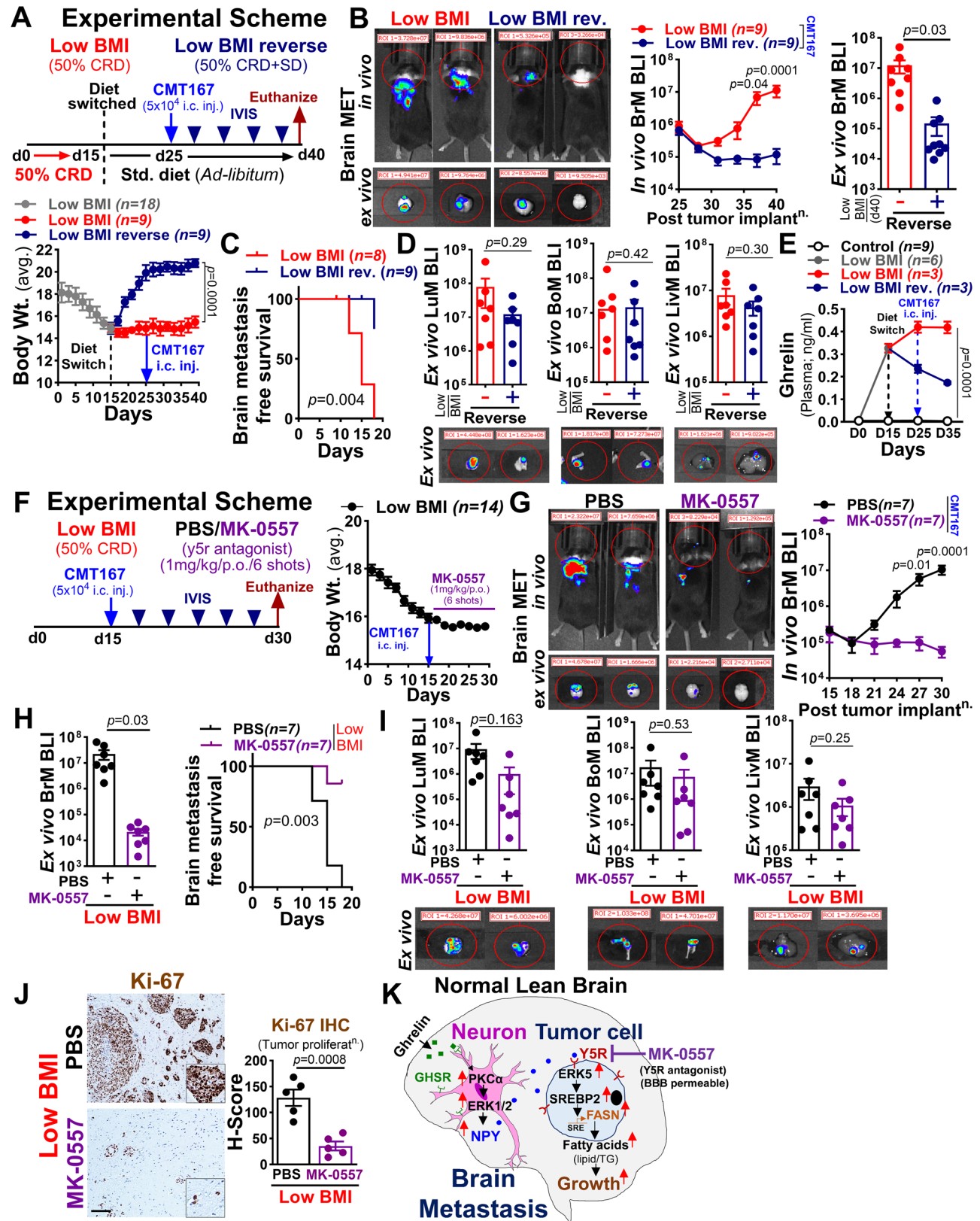

levels could serve as a promising prognostic biomarker for predicting an increased risk of brain metastasis in lung cancer patients with low BMI. Moreover, existing findings demonstrate the tumor-suppressive role of ghrelin in various cancers[104–106], consistent with our data. Furthermore, the specific activation of neurons by ghrelin compared to other brain cell types under low BMI conditions implicates its

functional role in supporting neuron survival during metabolic adaptation to calorie restriction, in agreement with previous studies demonstrating GHSR receptor-mediated enhanced neuronal activity in energy-deficit states[107–110]. Taken together, these observations highlight the intricate role of ghrelin in mediating the low-BMI phenotype and suggest a relationship between GHSR-activated neurons and the

**Fig. 5 | Low BMI reversibility or Y5R inhibition suppresses lung cancer brain metastasis. A** Top: Schematic of in vivo metastasis assay. Female C57BL/6 mice (5-6 weeks) were fed with a 50% CRD (12-hr/dark phase; 15 days), then switched to standard diet (10 days). On day 25, mice received CMT167 cells injection ($5 \times 10^4$; i.c.). Bottom: Average body weight (two-way ANOVA with Tukey's test). **B** Top left: Representative BLI images of brain metastasis at endpoint. Middle/Right: In vivo and ex vivo BLI quantification of brain metastasis in control (low BMI) and reversed low BMI mice ($n = 9$/group) [two-way ANOVA with Tukey's test (in vivo); unpaired two-tailed t-test (ex vivo)]. **C** Kaplan–Meier analysis of brain metastasis-free survival in control ($n = 8$) and reversed low BMI mice ($n = 9$) [log-rank (Mantel–Cox test)]. **D** Representative ex vivo BLI images and quantification of extracranial metastasis in control and reversed low BMI mice ($n = 7$/group; unpaired two-tailed t-test). **E** Serum ghrelin level in control ($n = 9$) and low BMI mice ($n = 6$) at indicated time points before/after diet switching (two-way ANOVA with Tukey's test). **F** Left: Schematic of in vivo metastasis assay. Female C57BL/6 mice (5-6 weeks) were fed with a 50% CRD (12-hr/dark phase; 15 days) before CMT167 cells injection ($5 \times 10^4$; i.c.). Mice were treated with PBS or MK-0557 (1 mg/kg/p.o./6-shots, $n = 7$/group). Right: Average body weight. **G** Left: Representative BLI images of brain metastasis at endpoint. Right: In vivo BLI quantification of brain metastasis in PBS or MK-0557 treated mice ($n = 7$/group; two-way ANOVA with Tukey's test). **H** Left: Ex vivo BLI quantification of brain metastasis in PBS or MK-0557 treated mice ($n = 7$/group; unpaired two-tailed t-test). Right: Kaplan–Meier analysis of brain metastasis-free survival in PBS or MK-0557 treated mice [$n = 7$/group; log-rank (Mantel–Cox test)]. **I** Representative ex vivo BLI images and quantification of extracranial metastasis in PBS or MK-0557 treated mice ($n = 7$/group; unpaired two-tailed t-test). **J** Left: Representative Ki-67 + IHC in metastatic brain tissue from PBS or MK-0557 treated mice [$n = 5$/group; Scale bar: 100 μm (Ki-67), insert (2x Zoom)]. Right: Dot plot showing Ki-67 quantification by H-score (unpaired two-tailed t-test). **K** Mechanism of low BMI-induced brain metastasis via neuronal NPY/Y5R signaling. All data are mean ± S.E.M. Source data are provided as a Source Data file.

development of metastatic disease in lung cancer patients with low BMI. Interestingly, accumulating evidence suggests that metformin, the anti-diabetic drug, directly inhibits ghrelin production and induces weight loss in both diabetic and non-diabetic individuals[111–115]. However, the impact of inherently lower ghrelin levels and the use of metformin in these individuals, particularly in the context of brain metastasis, is not yet fully understood. Notably, recent data indicate a significantly lower incidence of brain metastases in diabetic melanoma patients treated with metformin[116,117]. Furthermore, meta-analyses[118–121] have shown that diabetic patients receiving metformin exhibit a reduced risk of lung cancer compared to non-diabetic individuals. These findings suggest that metformin may play a confounding role in modulating lung cancer progression, highlighting the need for further mechanistic investigations.

Our study unveiled a significant increase in neuronal NPY expression and secretion in the brains of low BMI cancer-free subjects and tumor-free mice, as well as in ghrelin-pretreated primary neurons. Similarly, previous findings have shown a ~2-fold increase in brain NPY levels in ghrelin-activated neurons under fasting, in underweight anorexia patients, and during food-deprived conditions[122–125], which is significantly attenuated by pharmacological inhibition of ghrelin-mediated MAPK1 signaling in neurons[48,126], highlighting ghrelin's specific role in neuron activation and NPY secretion. Furthermore, our study demonstrated a significant increase in tumor growth upon NPY treatment, along with an increased brain metastatic burden in a low BMI setting, indicating its distinct role in lung cancer progression beyond regulating feeding behavior and energy homeostasis[127,128]. This was consistent with previous studies implicating dysfunction of the NPY system in various inflammatory pathologies, including cardiovascular disease, and neuroendocrine tumors through distinct signaling pathways via their receptors (Y1R, Y5R)[57,58,129–131]. Notably, we found significantly activated NPY receptor activity with substantially higher Y5R expression in matched brain metastatic tumors in contrast to primary tumors, or other NPY receptors or metastatic breast and melanoma cancers and this was significantly correlated with low BMI, resulting in lower brain metastasis-free survival in a Y5R/MAPK-dependent mechanism, indicating that MAPK1 serves as a key oncogenic driver contributing to the downstream effect of NPY in cancer cells. Nevertheless, our findings strongly suggest a critical role of the Y5R in mediating low BMI-induced neuronal NPY effects via the MAPK pathway in lung cancer progression and offer a potential therapeutic avenue to treat MAPK-driven cancers using ERK inhibitors, particularly in metastatic lung cancer patients with low BMI.

Recent evidence suggests that metabolic plasticity is a crucial determinant of tumor metastasis[64]. Differences in nutrient availability in brain tissue relative to other tissues can necessitate metabolic adaptation or competition by cancer cells for growth in the brain[65,132,133]. Furthermore, NPY has been shown to play a significant role in regulating

metabolic homeostasis and energy balance, as observed in condition such as obesity, hypertension, and atherosclerosis[134]. In this regard, we identified enrichment of activated signatures specific to fatty acid (FA) metabolism, biosynthesis, and oxidation in brain metastasis compared to primary tumors. Importantly, fatty acid metabolism has been shown to play a pro-tumoral role in cancer cells[135,136]. Although the underlying molecular mechanism(s) of this metabolic rewiring remains unclear, emerging evidence suggests that FA metabolism accelerates the incidence of brain tumor or metastasis, as both glycolysis and FAO pathways are essential for tumor metabolism, dictating tumor aggressiveness[65,137–141]. Such bioenergetic flexibility appears to fit well with the increased capacity of tumors for metastasis, in which fatty acids serve as an alternative critical energy resource to meet the high-fuel consumption of aggressively growing cancer cells. Intriguingly, our study revealed significantly upregulated FA metabolism, mainly FA-oxidation (FAO), in brain metastatic tumors under low BMI conditions, which was further enhanced upon NPY treatment in a Y5R-dependent manner. These findings are consistent with existing literature indicating tumor cells' dependency on elevated fatty acid metabolism for their survival in a non-conducive environment in metastatic brain tumors[65,138,140–142]. Moreover, fasting and calorie restriction regimens have been shown to increase FAO rates along with FA synthesis to maintain homeostasis[143,144], supporting our findings from clinical and pre-clinical models of brain metastasis with low BMI. Additionally, we observed elevated intracellular lipid droplets in Y5R[high]-tumor cells in correlation with increased levels of ERK, SREBP2, FASN and CPT1 in clinical samples of lung cancer brain metastasis and in patient-derived brain-metastatic xenografts. This effect was attenuated in the presence of an ERK5 inhibitor, indicating NPY-mediated active de novo lipogenesis in Y5R[high]-tumor cells through an ERK5/SREBP2-dependent mechanism, consistent with existing literature highlighting Y5R involvement in mediating lipogenesis[145]. Collectively, our results suggest that the energy derived from FAO contributes to supporting tumor growth during brain colonization under a low-BMI condition through the NPY/Y5R signaling axis.

By utilizing the highly selective, orally active, and blood-brain barrier (BBB) permeable Y5R antagonist, MK-0557[69,70], we found that a low dose (1 mg/kg) effectively abrogated low BMI-induced brain metastasis and significantly increased brain-metastasis free survival in vivo by blocking NPY/Y5R axis. Importantly, MK-0557 demonstrated this efficacy without affecting overall health or body weight in low BMI mice and intriguingly suppressed cancer cell viability and growth only at higher doses. However, its impact on body weight in normal mice remains poorly understood, as most studies on Y5R inhibition have primarily focused on obese or high-fat diet models[69,146,147], necessitating further investigation. Notably, MK-0557 treatment resulted in a significant reduction in intracellular lipid droplet formation within brain metastatic lesions. Considering its selective in vivo inhibitory effect on the Y5R, along with its relatively low toxicity and BBB

permeability, MK-0557 emerges as a potential therapeutic agent to suppress the onset of brain metastasis, particularly in lung cancer patients with low BMI. Our study provides insights into the mechanisms underlying low BMI-associated alterations in the interplay between tumor cells and brain cells. Specifically, we have elucidated the role of neuronal NPY in driving brain metastasis and highlighted its functional significance in regulating cancer cell survival and growth by reprogramming their metabolism. Moreover, the identification of elevated plasma ghrelin levels in individuals with low BMI holds significant clinical importance, particularly as a reliable prognostic marker for patients with lung cancer and low BMI. Additionally, the recognition of tumor metabolic dependency on fatty acid metabolism unveils a promising pharmacological avenue to effectively suppress NPY-activated cancers in the brain. Furthermore, the exploration of prospective dietary interventions and the utilization of the Y5R antagonist presents promising clinical translational applications for the treatment of lung cancer brain metastasis in patients with low BMI.

## Methods

### Human samples
Formalin-fixed paraffin-embedded (FFPE) tumor tissues from lung cancer brain metastatic patients with a BMI history were obtained from the Tumor Tissue and Pathology Shared Resource at Wake Forest Baptist Comprehensive Cancer Center. All samples were deidentified and collected in accordance with the Wake Forest School of Medicine IRB-approved protocol (IRB00107587). Informed consent was obtained from all participants. Demographic details, including BMI, age, gender, race, and smoking status were provided in the source data file.

### Cell culture and reagents
Human lung cancer brain-tropic cell lines H2030BrM and PC9BrM (derived from in vivo selection of H2030, PC9 cell line harboring the *Kras* mutation)[148] were a kind gift from Dr. Massague (Memorial Sloan-Kettering Cancer Center). Mouse lung carcinoma cell lines, LL/2 and CMT167, were purchased from American Type Culture Collection (ATCC) and MilliporeSigma. iPSC-derived GABAergic human primary neurons were purchased from BrainXell and maintained in neurobasal medium (ThermoFisher), supplemented with GlutaMAX (ThermoFisher), pen/strep (ThermoFisher), B-27 supplement (ThermoFisher), N-2 supplement (ThermoFisher), and BDNF (10 ng/mL; ThermoFisher). E6/E7/hTERT immortalized human astrocyte (UC1) were a kind gift from Dr Russell Piper (University of California-San Francisco) and were cultured in DMEM with 10% FBS. Human microglial cell line HMC3 was purchased from ATCC, authenticated by ATCC and cultured in DMEM/F12 medium supplemented with 5% FBS. All cells were cultured in their respective medium supplemented with 10% FBS, streptomycin (100 mg/ml), and penicillin (100 units/ml). All cells were grown at 37 °C in a 5% $CO_2$ atmosphere and were routinely tested for the absence of Mycoplasma.

### Animal experiments
All animal experiments were done in accordance with the U.S. National Institutes of Health Animal Protection Guidelines, and the protocol was approved by the Wake Forest Baptist Health Institutional Animal Care and Use Committee. All mice (C57BL/6; female, 5-6 week) were housed individually and were maintained in a specific pathogen free unit on a 12:12 light: dark cycle with either *ad-libitum* standard diet (5P00 - Prolab® RMH 3000 from LabDiet) or 50% calorie restriction diet (D19021107 from ResearchDiets) with time-restricted feeding and access to water. The animal rooms were provided with 100% fresh, HEPA filtered air at 10-15 air changes per hour. Room temperatures were controlled by reheat units within each room, and were maintained within the range of 70 °F ± 2° F. The humidity levels were controlled and maintained between 30-70%. Sample size was calculated based on previous experiments[149,150]. Using power analysis, the anticipated means and standard error of means were included to determine the sample size that was expected to yield a power of approximately 80 percent using a *p*-value of 0.05. All mice were age- and sex-matched and randomized into different experimental groups.

To examine the effect of BMI, we established a low-BMI mouse model (Supplementary Table 1) through a regimen of 50% calorie restriction combined with time-restricted feeding (mimicking human low BMI condition) and examined its role in experimental metastatic models by implanting luciferase-labeled CMT167 and LL2 mouse lung cancer cells in syngeneic C57BL/6 mice via intracardiac injection (i.c.) followed by monitoring the tumor metastatic burden by bioluminescence imaging (BLI). All mice were randomized before the experiment, blindly selected before injection. To confirm successful injection, whole-body bioluminescence imaging (BLI) was immediately performed using the IVIS imaging system. Tumor growth reaching a radiance of $10^8$ photons/sec/cm$^2$/sr was considered the humane endpoint. Tumor progression was monitored twice weekly using BLI. At the endpoint, mice were sacrificed, and the whole brain, lung, liver and bones were removed, and the presence of metastatic lesions on these organs was evaluated by ex vivo BLI imaging. The BLI images were analyzed using the Xenogen IVIS imaging system (Caliper Life Science).

### Bioluminescence imaging (BLI)
For in vivo monitoring of tumor growth, bioluminescence imaging was performed using an IVIS imaging system (Xenogen). Animals were placed under 1-4% isoflurane anesthesia and injected with luciferin substrate. Images were then acquired and analyzed using the IVIS imaging system (Xenogen). Baseline bioluminescence was used to randomize animals by a blinded investigator.

### Histological staining
Hematoxylin and eosin (H&E) staining was performed on formalin-fixed paraffin-embedded (FFPE) tumor tissues using standard protocols. Briefly, tissue was fixed in 10% formalin overnight, dehydrated, embedded in paraffin, and sectioned at 4−8 μm thicknesses using a microtome and processed for H&E staining. For immunohistochemistry (IHC), paraffin-embedded sections were de-paraffinized, rehydrated, and heated at 100 °C for 20 min in Tris-buffered saline (pH 9.0) for antigen exposure. The sections were treated with 3% $H_2O_2$ to block endogenous peroxidase activity, followed by incubation in 5% BSA solution for 30 min. The slides were then incubated with primary antibody, Ki-67 (Cell Signaling Technology, clone 24E10, #3195, 1:200) for 16 h at 4 °C. After washing in PBS/0.1% Tween-20, the sections were incubated with anti-rabbit secondary antibody (Bio-Rad, #1706515, 1:500). The sections were washed extensively, treated with DAB substrate chromogen solution, and counterstained with hematoxylin. For the negative control, the rabbit IgG isotype control (Invitrogen, 02-6102, 1:250) was used in place of primary antibody. The expression of the protein of interest was evaluated by H scoring, which was calculated as the sum of the frequency and intensity (0: none; 1: weak; 2: moderate; 3: strong). The final score = 1 × (% of 1 + cells) + 2 × (% of 2 + cells) + 3 × (% of 3 + cells). Image quantification was done using ImageJ software. IHC scores were determined based on concordance between two independent reviewers.

### Quantitative real-time PCR (RT-qPCR)
Total RNAs were extracted from indicated cells or FFPE sections using either the Direct-zol RNA kit (Zymo research) or the RNeasy FFPE kit (Qiagen), according to the manufacturer's instructions. The concentration of RNA was quantified using a NanoDrop 2000 UV-spectrophotometer (Thermo Scientific) and reverse transcribed into complementary DNA (cDNA) using the Reverse Transcription Supermix kit (Bio-Rad, USA). The cDNA was then amplified using a pair of forward and reverse primers for the genes listed in Supplementary

Table 3. Results were normalized to the housekeeping gene β-Actin. The thermal cycling conditions consisted of an initial denaturation step at 95 °C for 1 min., followed by 35 PCR cycles with the following profile: 94 °C for 30 s, 62 °C for 30 s, and 72 °C for 30 s, using the Bio-Rad CFX connect.

## Western blotting

Western blotting was performed using standard method. Briefly, total protein from cells was extracted using radio-immunoprecipitation assay buffer with Halt protease inhibitor cocktail (Thermo Fisher Scientific). Plasma membrane protein was extracted by the Plasma Membrane Protein Extraction Kit (Abcam). Protein concentration was measured using a protein assay (Bio-Rad). Equal amounts of protein were resolved by 10% SDS–polyacrylamide gel electrophoresis, followed by electro-transfer to nitrocellulose membranes. Primary antibodies used included PKCα (1:1000, Cell Signaling, #59754 T), pPKCα (1:1000, Cell Signaling, #44-962 G), ERK5 (1:1000, Cell Signaling, #D23E9), pERK5 (1:10000, Thermo-Fisher, #44-612 G), FoxO1 (1:1000, Cell Signaling, #2880 T), AMPKα (1:1000, cell Signaling, #2532S), pAMPKα (1:1000, cell Signaling, #2535 T), SirT1 (1:1000, cell Signaling, #8469 T), p53 (1:1000, cell Signaling, #2527 T), pCREB (1:1000, Cell Signaling, #9198 T), pSTAT3 (1:1000, Cell Signaling, #9145S), PPARγ (1:1000, Cell Signaling, #2443), FASN (1:1000, cell Signaling, #3180 T), SREBP2 (1:500, Novus Biologicals, #NB100-74543), pSREBP2 (1:500, ThermoFisher, #PA5-106042), NPY (1:500, Cell Signaling, #11976 T), NPY5R (1:500, ThermoFisher, #PA5-106850), BSX (1:100, MyBioSource.com, #MBS9609602) and GAPDH (1:10000, Cell Signaling, #5174 T). HRP-conjugated anti-mouse IgG (1:5000, Cell Signaling, #7076) or anti-rabbit IgG (1:5000, Bio-Rad, #1706515) were used as secondary antibodies. Bound antibodies were detected using an enhanced chemiluminescence detection kit (GE Healthcare Lifescience). Band intensities were measured using an Amersham Imager 600 and analyzed using ImageJ (version 1.53c http://rsb.info.nih.gov/ij/).

## ELISA

Ghrelin and NPY levels in plasma and conditioned media were measured using the mouse Ghrelin and human NPY ELISA Kits according to the manufacturer's instructions (MyBioSource; RayBiotech). A standard curve was generated using known concentrations of ghrelin and NPY provided by the kit, and the concentration of each sample was interpolated from it. All concentration was determined at a 450 nm wavelength using an EMax Plus microplate reader (Molecular Devices, CA).

## Generation of conditioned medium

Conditioned medium (CM) was generated from cultured human microglia, astrocytes lines (HMC3, UC1), followed by centrifuge at 300 g for 10 min to remove cells and debris, then stored at -80°C. CM from human primary neurons was prepared by seeding 2x10⁶ cells with or without recombinant ghrelin (0.01-0.25 nM) or recombinant neuropeptide (10-100 nM) in culture plates coated with poly-L-lysine (Sigma). After overnight incubation, cells were washed with PBS and further incubated in serum-free neuron-specific media for 24 h. The harvested CM was then centrifuged to remove cell debris (300 g/10 min) and stored at -80°C for further analysis.

## MTS assay

The 3-(4,5-dimethylthiazol-2-yl)-5-(3-carboxymethoxyphenyl)-2-(4-sulfophenyl)-2H-tetrazolium (MTS Promega) assay was used to determine cells viability with indicated drugs in a time- and dose-dependent manner. Briefly, cancer cells were seeded into 96-well plates and incubated overnight at 37 °C. The cells were then treated with indicated recombinant proteins or brain cell-specific CM in dose- and time-dependent manner. Thereafter, MTS solution was added to each well,

and the plates were incubated for 4 h at 37 °C. The media was then removed, and 150 µl DMSO was added to dissolve the formazan crystals. Absorbance was measured at 490 nm using the EMax Plus microplate reader (Molecular Devices, CA). Four to five replicate wells were included in each analysis, and at least three independent experiments were conducted.

## BrdU incorporation assay

Cancer cells were treated with or without recombinant NPY or ghrelin with the desired concentration as indicated in the figure. BrdU incorporation assay was performed using FITC BrdU Flow kit (BD Biosciences) according to the manufacturer's instructions. Briefly, cells were pulsed with 10 µM BrdU and stained with anti-BrdU antibody following DNA digestion. Cells were then stained with nuclear stain 7-AAD, and data was recorded by flow cytometry using the FACS Canto II flow cytometer and FACSDiva software (BD Biosciences) and analyzed by FlowJO. The gating strategy used for flow cytometric analysis of BrdU incorporation is provided in Supplementary fig. 8.

## Colony formation assay

Cells were treated with or without conditioned media (CM) derived from indicated brain cell types, or recombinant NPY, or ghrelin (to control media) and allowed to grow for 7 days. The media were replenished every 3rd day. On day 7, the colonies were fixed with ethanol for 30 min, stained with Crystal Violet (CV) for 20 min., and then the number of colonies was counted manually.

## Oil red O staining

Control and treated cells were washed and fixed in 4% paraformaldehyde at room temperature for 1 h. After washing three times with 60% isopropanol, the cells were stained with 60% filtered Oil Red O working solution (vol/vol in distilled water) prepared from Oil red O stock solution (#0843, ScienceCell Research) at room temperature for 15 min. The cells were then washed with ddH2O before imaging. To quantify lipid accumulation, Oil Red O-stained lipids were eluted in 100% isopropanol, and then the optical density (OD) was measured at 500 nm.

## Lipid droplet staining

For the lipid assay, 610 dye (#70069, Biotium) was used. Cells were incubated directly with LipidSpot™ 610 dye at a 1:1000 dilution for 1 h, protected from light in the incubator. Images were captured using the Keyence fluorescence microscope (BZ-X710).

## Fatty Acid β-Oxidation (FAO) driven Oxygen Consumption Rate (OCR)

FAO measurement was done by following the manufacturer's instruction of Fatty Acid Oxidation Assay Kit from (Abcam, ab222944). Briefly, control or y5r-KO CMT167 cancer cells were seeded per well in 96-well plates and cultured overnight. Then the cells were washed twice with 100 µl prewarmed FA-Free medium followed by adding 90 µl pre-warmed FA-Measurement medium. The wells without cells were used as signal control. A total of 85 µl of FA-Free Measurement medium was added to the wells, and 5 µl of BSA control were included as the FA-free control. All wells except the blank control had 10 µl Extracellular O₂ Consumption Reagent added. The FAO activator FCCP (0.625 µM) and inhibitor Etomoxir (40 µM) with or without FAO-conjugate (Oleate-BSA, 2:1) were added as positive and negative controls. Then the wells were sealed with 100 µl pre-warmed mineral oil, and the FAO was measured using the condition as Extracellular Oxygen Consumption by measuring fluorescence using Tecan microplate reader.

## Extracellular acidification rate (ECAR)

For ECAR assay (Abcam, ab197244), control or y5r-KO CMT167 cancer cells were cultured for 48 h, and culture media was changed to the

glycolysis buffer containing probe and the rate of glycolysis was determined using Glycolysis Assay Kit (Abcam, US) following manufacturers recommended using Tecan microplate reader.

### CRISPR/Cas9 based gene knockout (KO)

Npy5r knockout was performed using the CRISPR/Cas9 system. Briefly, CMT167 cancer cells were transfected with Cas9 protein V2 (Thermo-Fisher) and two guide RNA sequence (sgRNA: GUA-GUCCUCCCAGGCAGGG; UACUGCUGCCAGUCAGAACA) targeting the y5R sequence. Single-cell clones were isolated through limiting dilution following Synthego's protocol, and successful knockout was confirmed by western blot analysis.

### Plasmids

Expression plasmids for mouse Srebp2 and pCI-HA-Erk5-FL were obtained from Addgene. Each plasmid (1 μg) was independently transfected into LL2 lung cancer cells via either by lentiviral method or Lipofectamine 3000 (Invitrogen). Following transfection, cells were treated with or without recombinant Npy (100 nM) in presence or absence of Erk5 or Srebp inhibitors (Selleckhem). After 24 h, western blotting was conducted using the standard protocol described in western blotting section, employing respective primary and secondary antibodies.

### MK-0557 (NPY5R antagonist) dosing

MK-0557 was purchased from MedChemExpress (cat. no. HY-15411) and reconstituted to a concentration of 1 mg/ml MK-0557 according to the manufacturer's instructions. Mice received 50–100 μl of vehicle or 1 mg/kg MK-0557 via oral gavage (p.o.) immediately after intracardiac injection with $5 \times 10^4$ CMT167-luc cells for the indicated duration.

### RNA-sequencing data from published studies

RNA-sequencing data from lung cancer brain metastasis patient cohorts (GSE200563, GSE123902)[55], BrM-xenograft (GSE115699)[151], were analyzed by downloading data from their respective source file or utilizing supporting interactive web portals (http://bmxexplorer. gotdns.org/) and discovery platforms such as Rosalind (https://www. rosalind.bio/). The data were further processed for Gene Set Enrichment Analysis (GSEA) to investigate oncogenic and metabolic pathways. For overall or organ-specific free survival (OS), samples were either stratified by high or low expression of the gene of interest, or by BMI status and were presented as Kaplan–Meier plots and tested for significance using log-rank tests as previously described[152].

### Gene set enrichment analysis

GSEA was performed by generating the Gene MatriX file (.gmx) using published oncogenic signatures. The Gene Cluster Text file (.gct) was generated from the GEO database (GSE200563; GSE123902) either by separating primary and brain metastatic tumors or by stratifying brain metastatic tumors based on high or low expression of the gene of interest (Y5R). The Categorical class file (.cls) was generated based on the neuropeptide receptor activation scores and we used GPL21697, GPL16791 as the chip platform.

### ALT/AST Activity Assay

The ALT (alanine aminotransferase) activity was detected using ALT Activity Assay kit (Sigma-Aldrich, USA) according to the product information sheet. Briefly, 100 μL of the Master Reaction Mix was added to each of the standard, positive control, and test samples in each well of 96-well plates. After 2–3 min ($T_{initial}$), take the initial measurement of absorbance at 570 nm $(A_{570})_{initial}$. The plate was then incubated at 37 °C protected from light, and measurements were taken every 5 min until the highest value of the samples is greater than the highest value of the standards. The penultimate reading ($T_{final}$) was the

final measurement for calculating the enzyme activity $(A_{570})_{final}$. To calculate the ALT activity, the background was corrected by subtracting the value obtained for the blank standard from all standard readings. The amount of pyruvate generated (B) between $T_{initial}$ and $T_{final}$ was calculated by plotting the $\Delta A_{570} = (A_{570})_{final} - (A_{570})_{initial}$ to the to the pyruvate standard curve. AST (aspartate aminotransferase activity was measured with AST Activity Assay kit (Sigma-Aldrich, USA) according to product information sheet. Briefly, 100 μL of the Reaction Mix was added to each of the well. After 2–3 min ($T_{initial}$) incubation at 37 ˚C, initial measurement of absorbance at 450 nm $(A_{450})_{initial}$ was taken. The measurements were then taken every 5 min until the value of the most active sample is greater than the value of the highest standard. The penultimate reading ($T_{final}$) was the final measurement for calculating the enzyme activity $(A_{450})_{final}$. To calculate the ALT activity, background was corrected by subtracting the value obtained for the blank standard from all standard readings. The amount of glutamate generated (B) between $T_{initial}$ and $T_{final}$ was calculated by plotting the $\Delta A_{450} = (A_{570})_{final} - (A_{570})_{initial}$ to the glutamate standard curve. Both ALT, AST activity was calculated as B × Sample Dilution Factor/ (($T_{final} - T_{initial}$) × Sample volume).

### Statistical analysis and reproducibility

The number of mice used in each experimental group was determined by power analysis and on the basis of prior experiences. All the statistical analysis was performed using the Prism 10.0 (GraphPad). Data are expressed as mean ± S.E.M. The $P$ value was calculated using an unpaired Student's t-test or ANOVA, with corresponding numbers (n) as indicated in the figures and their respective figure legends. Gene correlation analysis was calculated by linear regression analysis. Kaplan–Meier survival analysis was calculated using the log-rank (Mantel–Cox) test. Representative experiments were repeated independently at least three times with similar results. Significance between each group is represented as *$P < 0.05$, **$P < 0.01$, ***$P < 0.001$, ****$P < 0.0001$, unless otherwise indicated.

### Reporting summary

Further information on research design is available in the Nature Portfolio Reporting Summary linked to this article.

## Data availability

Previously published micro-array, RNA-seq. and proteomic data that were reanalyzed in this study are available in GEO under accession codes GSE200563, GSE123902, GSE48433, GSE14020, GSE199089, GSE50493 or at PRIDE - Proteomics Identification Database under accession codes PXD027259. Human protein atlas, GTEx database, Kaplan–Meier survival analysis, TCGA lung, Gene Ontology referenced during the study are available in a public repository from https://www. proteinatlas.org/; https://www.gtexportal.org/home/; http://kmplot. com/analysis/index.php?p=service; http://geneontology.org/ https:// lce.biohpc.swmed.edu/lungcancer/; website. Source data are provided as a Source Data file. Source data are provided with this paper.

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

## Acknowledgements

This work was supported by NIH grant R01CA205067, W81XWH2110075 from Department of Defense (KW) and HT9425-24-1-0144 from Department of Defense (AT). This study utilized various Core Facilities and Departmental Shared Equipment resources, including the Cellular Imaging Shared Resource, Tumor Tissue and Pathology Shared Resource, Cell and Viral Vector Core Laboratory, and Flow Cytometry Shared Resource, all supported by the Comprehensive Cancer Center of Wake Forest University NCI, National Institutes of Health Grant (P30CA012197). We also thank Prof. Alessio Cortellini (Imperial College London, UK), Prof. Aoife Ryan (University College Cork, New York), and Prof. Marco Siringo (University of Rome, Italy) for kindly providing the materials used in this work.

## Author contributions

Conception and design: A.T., K.W. Development of methodology: A.T., K.W. Acquisition of data (provided animals, acquired and managed patient specimens, provided facilities, etc.): A.T., S.Y.W., J.W.K., R.P.D., K.W., E.C.S., G.L.B., K.W. Analysis and interpretation of data (e.g., statistical analysis, biostatistics, computational analysis): A.T., J.W.K., K.W. Writing, review, and/or revision of the manuscript: A.T., E.C.S., K.W. Administrative, technical, or material support: A.T., K.W. Study supervision: K.W.

## Competing interests

The authors declare no competing interests.
