## [Transparent Peer Review file · Nature Communications]

Ghrelin-induced neuronal NPY promotes brain metastasis in lung cancer patients with low BMI

Corresponding Author: Dr Kounosuke Watabe

Version 0:

Reviewer comments:

Reviewer #1

(Remarks to the Author)

The manuscript by Tyagi et al. reports interesting and novel findings mechanistically linking low body mass index (BMI) with an increased risk of brain metastasis in patients with lung cancer; the association that has been indicated by epidemiological data and may have future clinical implications. The authors report that low BMI leads to increased systemic levels of ghrelin, which in turn stimulates synthesis of neuropeptide Y (NPY) in the brain. NPY, acting via its Y5 receptor (Y5R), is a growth factor for lung cancer cells. While the authors provide compelling data in vitro and in vivo, some of the claims remain unsupported or need more in-depth analysis. The major points are as follows:

- 1) To directly prove that low BMI stimulates lung cancer brain metastasis the authors should treat mice on normal diet with ghrelin or inhibit ghrelin in mice with low BMI and test the impact of these interventions on the metastatic pattern.
- 2) To prove directly that ghrelin-induced increase in lung cancer cell proliferation is mediated by NPY, the authors should:
 - a) Show that ghrelin stimulation results in NPY release from neurons. In Fig 2F-G the authors are showing an increase in ghrelin mRNA levels upon treatment with ghrelin and based on this assume that NPY is being secreted. However, NPY release from neurons is tightly controlled. Thus, to prove that the observed increase in NPY mRNA translates to its secretion, the authors should perform NPY ELISA in conditioned medium from ghrelin-treated neurons.
 - b) Use the conditioned medium from ghrelin-treated neurons to stimulate lung cancer cell proliferation in the presence of NPY receptor antagonists.
- 3) The assessment of the role the NPY plays in lung cancer brain metastasis is limited to cell proliferation, while metastasis also involves other processes, such as affinity to the brain tissue, migration, invasiveness and survival. Moreover, other than the analysis of Ki67 staining in the tissue shown in Fig 1C, the proliferation is measured by MTS assay, which is based on metabolic activity and does not distinguish between the changes in cell death and proliferation. To make a claim that NPY stimulates proliferation, the authors should use a true proliferation assay (e.g. cell cycle analysis, DNA synthesis assay or proliferation marker expression).
- 4) There is no direct evidence for the causative role of changes in lipid metabolism in stimulating lung cancer cell proliferation. In fact, ERK activation itself can stimulate cell proliferation. To support this claim, the authors would need to block NPY-induced metabolic changes and show their effect on cell proliferation.

Minor points:

- 1) Fig 2B - it is not clear how the authors isolated specific types of brain cells; it is also not stated how the fresh brain samples, particularly human, were obtained. The figure legend suggests the use of cell lines for microglia and astrocytes, but there is no mention of them in the method section. Also, in the Results section related to this panel, the authors state that the expression was the highest in neurons, but no statistical analysis is provided to support this.
- 2) Fig 2C - it is not clear what type of primary neurons were cultured to collect the conditioned medium in this and subsequent experiments. The label of Y axis for the colony formation is misleading (100 cells/well).
- 3) Fig 2H is missing non-treated control.
- 4) Fig 2J – the Western blot shows NPY expression, but omits molecular weights of the detected bands. The mature form of NPY is a very small protein (approx. 4kDa), it is secreted from the cells and difficult to detect by Western blot. However, NPY precursors – prepro-NPY and pro-NPY are much larger and easier to detect. The authors should clarify which form they are detecting and add molecular weight markers. Moreover, this and other Western blots in the manuscript should be quantified and the results shown as a graph with statistical analysis.

- 5) Fig 3A – the authors state that “lung cancer exhibited significantly higher expression of the NPY5R (Y5R) receptor compared to other NPY receptors”, however no statistical analysis was shown to support this claim.
- 6) Fig 3D – the up-regulation of Y5R expression should be also shown on protein level, as Y5R can be regulated post-transcriptionally.
- 7) Supplementary Fig 3E – in the left panel (proliferation) it is not clear while the WT cells do not respond to treatment but the Scr-KO cells do. In the right panel, all genotypes should be shown with and without NPY treatment.
- 8) Fig 3K – the graphs lack statistical analysis comparing all Scr-KO to Y5R-KO for Normal and low BMI, as shown in Fig 3I for brain metastasis. For example, it looks like there may be a significant difference in bone metastasis between Y5R+ and Y5R- cells.
- 9) Fig 4G – left panel (proliferation) – all genotypes should be shown with and without NPY treatment; right panel (Western blot) – how the authors explain higher baseline pERK and pSrebp2 levels in Y5R KO cells? Was this a consistently observed effect? As stated before, all Western blots should be quantified and the results shown as graphs with statistical analysis.
- 10) Fig 5 – the experiment with BMI reversal is not very informative, as the tumor cells are injected when the mice reach normal body weight. What would be more informative is to see if the growth of brain metastases slows down when the mice are back on normal diet after cancer cell injection.
- 11) Supplementary Fig 5B-C – the authors show decrease in viability of the lung cancer cells upon treatment with really high concentrations of the Y5R antagonist (100 μ M). However, these cells are not stimulated with NPY. Do lung cancer cells secrete their own NPY?
- 12) Supplementary Fig 5D – in addition to lipid accumulation analysis, the authors should evaluate other features of the metastatic tissue from mice treated with Y5R antagonist, such as proliferation and cell death levels.
- 13) Method section – the description of the brain cell isolation and colony formation assay is too superficial to understand the procedure.

General comments:

- 1) Overall, the language needs improvement. There are numerous grammatical errors (including the title) and many sentences are difficult to understand.
- 2) Figures are not well organized and some of them span two pages. The description of the panels (e.g. left, bottom) often does not match their actual locations in the figures.

Reviewer #2

(Remarks to the Author)

This manuscript investigates the molecular mechanisms under the low BMI and increased brain metastasis risk in lung cancer patients. The data show that low BMI increased ghrelin-induced secretion of neuronal NPY, which promotes metabolic reprogramming of tumors via Y5 receptors, making them more prone to fatty acid synthesis. Overall, this is an interesting study and the experimental approach is sound. However, some issues need to be addressed.

1. The canonical action of ghrelin on NPY neurons is based on p53/SIRT1 and AMPK/mTOR signaling pathways, at least at hypothalamic level. However, these molecular mechanisms, which have also a major (direct) impact on fatty acid metabolism (synthesis and oxidation; see below) have been totally ignored in this study, why? This should be explored, as well as the key transcriptional machinery modulating these effects, such as BSX, FoXO1 and CREB.
2. In keeping with the former point, the increased ghrelin tone in neurons is known to be associated with decreased acetyl-CoA carboxylase activity and decreased Srebp1 and Fasn expression, as well as FAS activity. The overall effect is a malonyl-CoA reduction, therefore leading to increased CPT1 activity and fatty acid oxidation. This is opposite to the observed effects, with increased Fasn expression and fatty acid accretion. I agree these are tumor cells, but still molecular details are demanding. Therefore, this should be investigated in depth and explained.
3. Y5 expression is increased upon NPY treatment. The molecular mechanism of this action needs to be investigated.
4. Line 94, I think it should read Fig. 1E instead of Fig. 1D.

Reviewer #3

(Remarks to the Author)

The inverse relationship between obesity and NSCLC outcome termed obesity paradox has been confounded by many factors including anti-diabetic drug metformin use in obese patients and BMI not being able to distinguish abdominal obesity. This is an active area of investigation. Tyagi et al performed a detailed characterization of low BMI-associated lung-to-brain metastasis and attributed it to increase in neuronal neuropeptide Y (NPY) secretion via ghrelin-GHSR receptor activation reprogramming the cancer cell energy metabolism. Increase in serum ghrelin in low BMI condition activates NPY which supports Y5R(high) tumor cells and prime the cells for brain metastasis.

The manuscript is well written and underlying biological findings are novel. Both in vitro and in vivo experiments are provided to validate BMI dependence. Npy5R knockout as well as pharmacologic inhibition using y5R antagonist show reduced brain metastasis under low BMI. There are several important weaknesses in the data and text, which if addressed, could significantly improve the study. This includes lack of rigor and/or clarity in the analysis of certain key pieces of data and methods (also described further under “Additional points”).

- Figure 1A: presents the data as % incidence, but there is no apparent consideration for time in the analysis. If incidence is the endpoint, then a Kaplan Meir analysis would be more appropriate.
- For all the in vivo analysis of brain metastasis, H&E images should be provided to corroborate key BLI based conclusions (e.g. Fig 1E, 3I, 5G).
- The manuscript relies heavily on GSE200563, which includes pairs of primary lung cancers and brain metastasis. However, in this prior study/dataset, the authors include separate gene expression data from the tumor stroma microenvironment as well as the tumor core. It is not clear how the present authors analyzed these data to reach their conclusions (e.g. in Fig. 3G).
- Line 575 authors mention “freshly isolated brain cells (human and mouse)” for conditioned medium in neuron culture. It is however not clear whether human/mouse neurons were used in experiments shown in figure 2B, 2C and 2D. How were fresh human brain cells isolated and cultured? What age were mouse neurons isolated? If mouse neurons were isolated from prenatal mice or day 0 pups, were the expression level for ghrelin-GHSR receptor and NPY like those of adult mouse neurons? If adult mice were used for neuron isolation, what was the BMI status for these mice?
- The impact of conditioned medium is shown in BrM cell lines in figure 2C. BrM lines are shown to have low GHSR expression in figure 2B. Since these cell lines have already been through the brain microenvironment once, it would be important to know the levels in parental lines.
- Line 423: The authors mention “Glycolysis is less important for GBM metabolism”. However, GBM tumors have significant up-regulation of both glucose and fatty acid oxidation pathways (<https://pubmed.ncbi.nlm.nih.gov/33854970/>) and this depends on aggressiveness in GBM. Please restructure the sentence to reflect that.
- There is also lack of careful consideration of causality and bias on metformin use. There is no discussion of metformin use in obese patients. Metformin directly inhibits ghrelin production and causes weight loss in both diabetic and non-diabetic individuals. What is the result of impact of inherent less ghrelin in these individuals, how can this be separated from additional impact of low BMI, and what portion of the phenotype is likely cell autonomous due to low expression of this receptor in high BMI individuals?

Additional points:

1. Line 159: Replace “we quired” with “we queried”.
2. Line 515: Replace “then imaged were evaluated by” with “then images were evaluated by”
3. Figure 1E: Do the ROIs used for all in-vivo analyses placed to reflect similar cranial area for all animals? Do the ROIs used for all ex-vivo analyses have the same dimensions?
4. Figure 3A: P values are not defined.
5. Figure 2C: Increased proliferation in neuron CM is only shown for H2030BrM. Was the same phenotype observed for other cell lines? Also, what is the time frame for the clonogenic experiments (endpoint and treatment interval). How is it possible to see an effect, if the treatments are presumably transient?
6. Figure 2D x-axis labeling is missing
7. Figure 4D: Western blot for total SREBP2 is missing.
8. Figure 4E: There is difference in intensity within the western blot band for pSREBP2. Did the authors corroborate the findings in a replicate?
9. Figure 4G: What is the explanation for reduced pErk in the y5rKO+rNPY treated relative to y5rKO unstimulated? A few more time points for this analysis might reveal the proper kinetics of this activation

Reviewer #4

(Remarks to the Author)

I co-reviewed this manuscript with one of the reviewers who provided the listed reports. This is part of the Nature Communications initiative to facilitate training in peer review and to provide appropriate recognition for Early Career Researchers who co-review manuscripts

Version 1:

Reviewer comments:

Reviewer #1

(Remarks to the Author)

The authors provided a significant amount of new data and successfully addressed the vast majority of previous concerns. These changes markedly improved the quality of the manuscript and supported the claims included in the paper. However, there are some additional changes that need to be made.

Comments regarding experiments:

1. Figure 2L – interpretation of the data presented in this panel is not correct. Typically, accumulation of cells in G1 phase indicates inhibition of cell proliferation rather than its stimulation. The results of the cell cycle analysis may depend on the time after stimulation. Perhaps the assay that measures cumulative DNA synthesis, such as BrdU or EdU incorporation, will give more reliable results.
2. Supplementary figure 4F is missing Scr KO and Y5R KO cell lines without NPY treatment. It would be interesting to know if Y5R KO affects basal cell proliferation.

Comments regarding figure organization and text:

1. The title requires revision. There is no such thing as “low BMI lung cancer”.
2. Upon addition of new data, the supplementary figures are not referenced in the text in order. Supplementary figure 2A and Supplementary figure 3 should be combined and presented as Supplementary figure 2, while current Supplementary Figure 2 B-K should become Supplementary figure 3.
3. Figure 2J-K – the labeling is confusing, as only panel J is labeled “H2030BrM”, while both panels J and K present data obtained using this cell line. Moreover, the difference between the proliferation assays shown in panels J and K is not stated in the legend (short term vs long term treatment).
4. Figure 2M – it is not mentioned which ERK is being detected. Similarly, in the text describing Supplementary figure 2K it should be mentioned that the inhibitor used in the experiment targets ERK1/2. Lastly, NPY at the molecular weight of 11kDa should be labeled as pro-NPY, since the molecular weight of mature NPY is lower.
5. Figure 3D – GSEA analysis shows enrichment of genes involved in “neuropeptide receptor activity”. However, the authors refer to it specifically as enrichment in NPY receptor activity. As there are many neuropeptides, it is not clear which interpretation is correct.
6. Lines 178-181 – the sentence describing Supplementary figure 4A is not clear
7. Lines 184-186 – the sentence describing Supplementary figure 4B is not clear
8. Line 221 and Figure 3 panel I – Yr5 – typo
9. Line 264 – the sentence is not clear. Please add “... in NPY-treated parental lung cancer cells”
10. Legend for Figure 5 – the descriptions of panels D and E are switched.
11. While describing the lack of the effect of MK-0557 on the mice body weight, the authors do not mention that the treatment was done in mice with already low body weight. Y5R inhibition may still affect weight of normal body weight mice. This should be discussed.
12. In the discussion section, the authors provide the full name of many terms abbreviated in earlier parts of the manuscript, such as NPY. This is not necessary.
13. Throughout the manuscript the authors use the phrase “Y5R receptor”. In the Y5R abbreviation, R stands for “receptor”. Thus “Y5R” should be sufficient.

Reviewer #2

(Remarks to the Author)

All my comments have been addressed. This is an interesting and technically sound manuscript.

Reviewer #3

(Remarks to the Author)

1. The authors addressed the question of neuron conditioned medium by providing the source of neurons and included it in the revised Methods section.
2. For the comment on GHSR expression data for parental cell lines, authors showed lower expression in parental as compared to neurons. But is it lower in BrM lines than the parental lines, which was the original comment. To address this, could the authors show BrM and parental lines on the same graph?
3. Line 423 in original manuscript has been satisfactorily modified based on current literature for GBM metabolism.
4. For metformin use, the authors have included relevant publications in the discussion. However, since metformin has anti-tumor effects in lung cancer (<https://doi.org/10.1016/j.jtho.2020.08.021>, <https://doi.org/10.18632/oncotarget.17066>), it would be important to include it as a confounding factor in context of cancer. Please include a sentence explaining role of metformin in lung cancer in the discussion.

All additional comments have been addressed.

Reviewer #4

(Remarks to the Author)

Version 2:

Reviewer comments:

Reviewer #1

(Remarks to the Author)

Thank you for the thorough response to the reviewers' comments. All my concerns have been addressed. The study is very well designed; the findings are well supported and impactful. Congratulations.

Point-by-point response to Reviewer's Comment's

We thank all the Reviewers for their thorough and constructive comments, which have enabled us to greatly improve the manuscript. As described below, we have now addressed all the critiques and suggestions from the reviewers and hope that the manuscript can be now considered for publication in *Nature Communications*. The responses appear in dark-red font. Changes in the manuscript have been highlighted to facilitate review.

Reviewer #1

The manuscript by Tyagi et al. reports interesting and novel findings mechanistically linking low body mass index (BMI) with an increased risk of brain metastasis in patients with lung cancer; the association that has been indicated by epidemiological data and may have future clinical implications. The authors report that low BMI leads to increased systemic levels of ghrelin, which in turn stimulates synthesis of neuropeptide Y (NPY) in the brain. NPY, acting via its Y5 receptor (Y5R), is a growth factor for lung cancer cells. While the authors provide compelling data in vitro and in vivo, some of the claims remain unsupported or need more in-depth analysis.

We thank the reviewer for their recognition of the interesting and novel aspects of our findings.

Major comments:

1. To directly prove that low BMI stimulates lung cancer brain metastasis the authors should treat mice on normal diet with ghrelin or inhibit ghrelin in mice with low BMI and test the impact of these interventions on the metastatic pattern?

Response: We appreciate the reviewer's comment. To further substantiate our findings, we have performed additional *in vivo* experiments to evaluate the metastatic burden under (i) ghrelin treatment in normal BMI mice and (ii) ghrelin receptor inhibition in low BMI mice. Our results demonstrate that normal BMI mice treated with ghrelin, as well as control low BMI mice, exhibit a significant increase in brain metastatic burden and reduced brain-metastasis free survival compared to PBS-treated normal BMI mice or low BMI mice receiving ghrelin receptor inhibition. On the other hand, these effects were not significant in extracranial metastatic sites. Altogether, these results strongly suggest that low BMI promotes lung cancer brain metastasis in a GHSR receptor-dependent manner. The new data are presented in the revised manuscript. (see revised Figure 2, panel B, Supplementary Figure 2, panel A and Supplementary Figure 3, panel A-C (also see figure below)).

Figure 2: Low BMI promote tumor growth via ghrelin activated neuronal NPY. B. Left panel: Schematic diagram of the metastasis assay. Immune-competent C57BL/6 mice were fed a normal (*ad-libitum*) diet and administration ghrelin (0.05mg/kg; i.p.; 5 shots) until day 10, followed by i.c. of CMT167 cells (5×10^4). Middle and right panel: *In vivo* and *ex vivo* quantification of brain metastasis in control ($n=6$) and ghrelin treated normal BMI mice ($n=8$) by BLI at the endpoint (two-way ANOVA with Tukey's multiple comparisons test (*in vivo*); unpaired two-tailed t-test (*ex vivo*)).

Supplementary Figure 2: Low BMI-induced ghrelin activates neuronal NPY and promotes tumor growth. A. Left panel: Brain metastasis-free survival of control and ghrelin-treated normal BMI mice ($n=6$ each) as analyzed by Kaplan–Meier analysis (log-rank [Mantel–Cox] test). Right panel: *Ex vivo* quantification of metastasis in lung, liver and bone in control and ghrelin-treated normal BMI mice by BLI at the endpoint ($n=6$ each, unpaired two-tailed t-test).

Supplementary Figure 3: Ghrelin receptor inhibition abrogates low BMI induced brain metastasis.

A. Upper left panel: Schematic of the metastasis assay. Immune-competent C57BL/6 mice were fed a 50% calorie restricted diet (12-hr in dark phase) for 15 days, followed by ghrelin receptor inhibitor administration by intraperitoneal injection (JMV2959; 3mg/kg; 5 shots) post-intracardiac injection of LL2 lung cancer cells (5×10^4). Lower left panel: Average body weight curve of each group throughout the study (two-way ANOVA with Tukey's multiple comparisons test). Upper right panel: *In vivo* and *ex vivo* quantification of brain metastasis in control ($n=4$) and ghrelin receptor inhibitor-treated ($n=6$) low BMI mice by BLI at the endpoint (two-way ANOVA with Tukey's multiple comparisons test (*in vivo*); unpaired two-tailed t-test (*ex vivo*)). **B.** Kaplan–Meier analysis of brain metastasis-free survival in control ($n=4$) and ghrelin receptor inhibitor-treated low BMI mice ($n=6$) [log-rank (Mantel–Cox test)]. **C.** *Ex vivo* quantification of metastasis in lung, liver and bone in control and ghrelin receptor inhibitor-treated low BMI mice ($n=4$ mice/group) by BLI at the endpoint (unpaired two-tailed t-test).

2. To prove directly that ghrelin-induced increase in lung cancer cell proliferation is mediated by NPY, the authors should:
- Show that ghrelin stimulation results in NPY release from neurons. In Fig 2F-G the authors are showing an increase in ghrelin mRNA levels upon treatment with ghrelin and based on this assume that NPY is being secreted. However, NPY release from neurons is tightly controlled. Thus, to prove that the observed increase in NPY mRNA translates to its secretion, the authors should perform NPY ELISA in conditioned medium from ghrelin-treated neurons.
 - Use the conditioned medium from ghrelin-treated neurons to stimulate lung cancer cell proliferation in the presence of NPY receptor antagonists

Response: We appreciate the reviewers' suggestion. To address these points, we have included the data of (i) neuronal NPY levels in the conditioned medium (CM) of human primary neurons treated with or without recombinant ghrelin and (ii) assessment of lung cancer cell proliferation in response to CM from ghrelin-treated neurons, with or without NPY receptor antagonists. We found significantly higher levels of NPY in the CM from ghrelin-treated neurons compared to control neurons. Furthermore, CM from ghrelin-treated neurons significantly stimulates lung cancer cell proliferation (H2030BrM), which is effectively inhibited by NPY receptor antagonists. These findings confirm that the ghrelin-induced cancer cell proliferation is mediated through NPY signaling (see new data in revised Figure 2, panel I and Supplementary Figure 6, panel G; also see figures below).

Figure 2: Low BMI promote tumor growth via ghrelin activated neuronal NPY. I. NPY protein levels in CM derived from ghrelin-treated (0.25nM) or untreated neurons were quantified by ELISA ($n = 3$ individual experiment/group, unpaired two-tailed t-test).

G H2030BrM cells

Supplementary Figure 6: Reversal of low BMI or Y5R inhibition attenuates brain metastasis. G. H2030BrM lung cancer cells were treated with or without conditioned medium derived from ghrelin-pretreated primary neurons, in the presence or absence of NPY receptor antagonist (MK-0557). Cell proliferation was assessed by MTS assay ($n=3$ individual experiments, unpaired two-tailed t-test).

- The assessment of the role the NPY plays in lung cancer brain metastasis is limited to cell proliferation, while metastasis also involves other processes, such as affinity to the brain tissue, migration, invasiveness and survival. Moreover, other than the analysis of Ki67 staining in the tissue shown in Fig 1C, the proliferation is measured by MTS assay, which is based on metabolic activity and does not distinguish between the changes in cell death and proliferation. To make a claim that NPY stimulates proliferation, the authors should use a true proliferation assay (e.g. cell cycle analysis, DNA synthesis assay or proliferation marker expression).

Response: We agree with the reviewer's observation and, in response, performed cell cycle analysis to assess the functional role of NPY on cancer cell proliferation. We found that NPY significantly enhances proliferation by promoting progression through the G1 and G2 phases of the cell cycle compared to controls. These findings further support the increased proliferation of cancer cells via the NPY/NPY5R signaling axis (see new data in revised Figure 2, panel L; also see figure below).

Figure 2: Low BMI promote tumor growth via ghrelin activated neuronal NPY. L. Left panel: Representative histogram of H2030BrM cells treated with or without recombinant NPY (100nM), analyzed for cell cycle distribution via flow cytometry (FACS) after propidium iodide (PI) staining. Right panel: Quantification of sub-G₁/G₀, G₁, S, and G₂ cell cycle fractions ($n = 3$ individual experiment, unpaired two-tailed t-test).

4. There is no direct evidence for the causative role of changes in lipid metabolism in stimulating lung cancer cell proliferation. In fact, ERK activation itself can stimulate cell proliferation. To support this claim, the authors would need to block NPY-induced metabolic changes and show their effect on cell proliferation.

Response: We appreciate the reviewer's insightful comment. To address the concern regarding the causative role of lipid metabolism in stimulating lung cancer cell proliferation, we performed additional experiments to block NPY-induced metabolic changes and assess their effects on cell proliferation. Specifically, we inhibited CPT1 using Etomoxir to block fatty acid oxidation (FAO). Briefly, for the cell proliferation assay, cancer cells were treated with a lower dose of the CPT1 inhibitor (40 μ M Etomoxir) to minimize toxicity, followed by incubation with or without recombinant NPY. Our results show that inhibition of CPT1 significantly reduced proliferation, suggesting that NPY-induced metabolic changes via the SREBP2/FASN/CPT1 axis play a key role in promoting cell proliferation (see new data in revised Supplementary Figure 5, panel E; also see figure below). These findings, consistent with our Y5R knockout metabolomic data, support the notion that metabolic changes contribute to the proliferative effects of NPY during the early stages of metastatic colonization in the brain.

Supplementary Figure 5: Neuronal NPY upregulated lipogenesis to promote tumor cell growth. E. H2030BrM lung cancer cells were treated with or without CPT1 inhibitor (Etomoxir) and examined for cell proliferation by MTS assay ($n=3$ individual experiments, unpaired two-tailed t-test).

Minor comments:

1. Fig 2B - t is not clear how the authors isolated specific types of brain cells; it is also not stated how the fresh brain samples, particularly human, were obtained. The figure legend suggests the use of cell lines for microglia and astrocytes, but there is no mention of them in the method section. Also, in the Results section related to this panel, the authors state that the expression was the highest in neurons, but no statistical analysis is provided to support this.

Response: We apologize for inadvertently providing incorrect description of the source of brain cell types and their derived conditioned media in the initial submission. For this study, we utilized commercially available iPSC-derived human primary neurons from BrainXell, human microglia (HMC3 cells) from ATCC, and E6/E7/hTERT immortalized human astrocytes (UC1), a kind gift from Dr. Russell Piper (UCSF). These cell types were cultured in their respective media, with or

without the indicated recombinant protein treatments, to generate all relevant data and associated conditioned media. We have now corrected this information in the Methods section of the revised manuscript. Additionally, as per the reviewer's suggestion, we have included statistical analysis comparing the expression levels between different brain cell types and between neurons and cancer cells (see revised Figure 2, panel C).

- 2C - it is not clear what type of primary neurons were cultured to collect the conditioned medium in this and subsequent experiments. The label of Y axis for the colony formation is misleading (100 cells/well).

Response: We apologize for inadvertently missing the description of the source of primary neurons in the initial submission. For this study, we utilized commercially available iPSC-derived human primary neurons from BrainXell to generate all relevant data and their associated conditioned media, which were cultured in their respective media with or without the indicated recombinant protein treatment. This information has now been corrected in the Methods section of the revised manuscript. Additionally, we have corrected the Y-axis labeling to eliminate any discrepancies (see revised Figure 2, panel D).

3. Fig 2H is missing non-treated control.

Response: We appreciate the reviewer's observation. In the revised version, we have repeated the experiments shown in Figure 2H using the indicated cancer cells, both with and without recombinant NPY, including an untreated control group (see new data in revised Figure 2, panel J; also see figure below).

Figure 2: Low BMI promote tumor growth via ghrelin activated neuronal NPY. J. H2030BrM lung cancer cells were treated with or without different dose of recombinant neuropeptide protein (NPY; 10nM-100nM) and examined for cell proliferation by MTS assay ($n = 3$ individual experiment, unpaired two-tailed t-test).

4. Fig 2J – the Western blot shows NPY expression but omits molecular weights of the detected bands. The mature form of NPY is a very small protein (approx. 4kDa), it is secreted from the cells and difficult to detect by Western blot. However, NPY precursors – prepro-NPY and pro-NPY are much larger and easier to detect. The authors should clarify which form they are detecting and add molecular weight markers. Moreover, this and other Western blots in the manuscript should be quantified and the results shown as a graph with statistical analysis.

Response: We acknowledge the reviewer's concern. We used the NPY antibody (Clone D75YA) from Cell Signaling (catalog #11976T), which detects both the mature (4 kDa) and precursor (11 kDa) forms of NPY. Additionally, in response to the reviewer's suggestion, we have included the

molecular weight markers for all western blot throughout the manuscript and provided the quantitative data as a graph with statistical analysis in the revised manuscript.

- Fig 3A – he authors state that “lung cancer exhibited significantly higher expression of the NPY5R (Y5R) receptor compared to other NPY receptors”, however no statistical analysis was shown to support this claim.

Response: Thank you for this valuable suggestion. In the revised version, we have included individual *p*-values for statistical comparisons between different NPY receptors in primary and brain metastatic clinical samples from lung, breast, and melanoma tumors. Additionally, we have highlighted the statistical analysis in the figure legend (see revised Figure 3, panel A).

- Fig 3D – the up-regulation of Y5R expression should be also shown on protein level, as Y5R can be regulated post-transcriptionally.

Response: As per reviewer’s recommendation, we have performed new western blot experiments to assess Y5R receptor expression in the indicated cancer cell lines treated with or without recombinant NPY. The new data have been moved to Supplementary Figure 4, panel D (also see figure below). Additionally, we have included quantitative analysis of the band intensities, presented as a graph with statistical comparisons, in the revised manuscript.

Supplementary Figure 4: Neuronal NPY enhances brain metastatic growth via Y5R receptor signaling. D. Left panel: Western blot analysis of Y5R receptor expression in mouse and human lung cancer cell lines (LL/2, CMT167, H2030BrM, PC9BrM) treated with or without recombinant NPY. GAPDH was used as a normalization control (*n*=3 individual experiments). Right panel: Quantitative analysis of band intensities using ImageJ (unpaired two-tailed t-test).

- Supplementary Fig 3E – in the left panel (proliferation) it is not clear while the WT cells do not respond to treatment but the Scr-KO cells do. In the right panel, all genotypes should be shown with and without NPY treatment.

Response: We appreciate the reviewer’s comment. The reduced proliferation observed in wild-type (WT) cancer cells compared to scrambled KO cells (Scr-KO) is due to the absence of recombinant NPY treatment. To address any potential discrepancies, we have repeated the cell proliferation and colony formation experiment using WT, Scr-KO and Y5R-KO cancer cells with or without recombinant NPY treatment. The revised data have been moved to Supplementary Figure 4, panel E (also see figure below).

Supplementary Figure 4: Neuronal NPY enhances brain metastatic growth via Y5R receptor signaling. F. Cells from panel D were treated with or without recombinant NPY protein (100nM) and examined for cell proliferation by MTS assay (left panel) and growth by colony formation assay (right panel) ($n=3$ individual experiments, unpaired two-tailed t-test).

8. Fig 3K – the graphs lack statistical analysis comparing all Scr-KO to Y5R-KO for Normal and low BMI, as shown in Fig 3I for brain metastasis. For example, it looks like there may be a significant difference in bone metastasis between Y5R+ and Y5R- cells.

Response: In the revised manuscript, we have included individual p -values for statistical comparisons between Scr-KO and Y5R-KO across all extracranial metastatic organs under both normal and low BMI conditions to rule out any discrepancy (see revised Figure 3, panel J).

9. Fig 4G – left panel (proliferation) – all genotypes should be shown with and without NPY treatment; right panel (Western blot) – how the authors explain higher baseline pERK and pSrebp2 levels in Y5R KO cells? Was this a consistently observed effect? As stated before, all Western blots should be quantified and the results shown as graphs with statistical analysis.

Response: We appreciate the reviewer's comment and would like to point out that Figure 4G represent qRT-PCR data. Following the reviewer's suggestion, we have repeated the entire experiment with all genotypes with or without recombinant NPY treatment (see new data in revised Figure 4G; also see figure below).

Figure 4. Low BMI-induced neuronal NPY upregulates lipogenesis via Y5R/SREBP2/FASN signaling axis to fuel tumor cell growth. G. Expression of Srebp2, Fasn and Cpt1a in Scr-KO and y5r-KO CMT167 cells treated with or without recombinant NPY (100nM), assessed by qRT-PCR. β -Actin was used as a normalization control ($n=3$ individual experiments, unpaired two-tailed t-test).

Furthermore, in the initial submission, we inadvertently used primary tumor core data for the correlation plots in Figure 3G and performed incorrect western blots for the indicated genes in Figure 4E and 4G. We apologize for this oversight. This has been rectified by using the appropriate brain metastatic (LB) tumor core data and re-plotting the correlation plots (now Figure 3, panel F). Additionally, we repeated the western blot experiments using either NPY- or ERK5 inhibitor-treated ScrKO and Y5RKO cancer cells. Our revised data demonstrate significantly higher levels of pERK5 and pSREBP2 in ScrKO cells following NPY stimulation compared to unstimulated control or Y5RKO cells, which showed only basal expression for both proteins. Notably, basal ERK5 expression was suppressed in both cell types upon ERK5 inhibitor treatment, irrespective of NPY stimulation (see revised 4H; also see figure below). This unexpected reduction in basal ERK5 levels may be result from the activation of stress-related pathways or from alterations in post-translational modifications. The corrected data, along with quantitative analysis and statistical comparisons, have been included in the revised manuscript.

Figure 4. Low BMI-induced neuronal NPY upregulates lipogenesis via Y5R/SREBP2/FASN signaling axis to fuel tumor cell growth. H. Upper panel: Cells from panel G were further immunoblotted for y5r, activated ERK5 and SREBP2 in the presence or absence of recombinant NPY protein or ERK5 inhibitor (100 nM each). GAPDH was used as a normalization control ($n=3$ individual experiments, unpaired two-tailed t-test). Lower panel: Quantitative analysis of band intensities using ImageJ.

10. Fig 5 – the experiment with BMI reversal is not very informative, as the tumor cells are injected when the mice reach normal body weight. What would be more informative is to see if the growth of brain metastases slows down when the mice are back on normal diet after cancer cell injection.

Response: We thank the reviewer for this insightful comment. In response, we conducted new *in vivo* experiment involving dietary intervention post-tumor implantation to assess brain metastasis. Consistent with the results from dietary intervention before tumor cell implantation, we found that dietary intervention after tumor cell implantation significantly reduced brain metastasis by over 20-fold and increased body weight compared to the control group, which remained on a low-BMI regimen. This intervention also led to a significant improvement in brain metastasis-free survival relative to the control group. Importantly, mice subject to dietary intervention did not show a significant reduction in extracranial metastases (lung, bone, and liver) compared to brain metastasis, consistent with our clinical observation (see new data in Supplementary Figure 6 panel A-C; also see figure below). These findings underscore the potential of low BMI as a modifiable risk factor for abrogating brain metastasis in lung cancer.

Supplementary Figure 6: Reversal of low BMI or Y5R inhibition attenuates brain metastasis. A. Upper panel: Schematic of the metastasis assay. Immune-competent C57BL/6 mice were fed a 50% calorie restricted diet (12-hr in dark phase) for 15 days, followed by a diet switch to a standard diet (*ad-libitum*) post-implantation of CMT167 cells (5x10⁴) via intracardiac injection. Lower panel: Average body weight curve of each group during the study period (two-way ANOVA with Tukey's multiple comparisons test). **B.** Left and middle panel: *In vivo* and *ex vivo* quantification of brain metastasis in control and reversed low BMI mice (n=6 each) by BLI at the endpoint (two-way ANOVA with Tukey's multiple comparisons test (*in vivo*); unpaired two-tailed t-test (*ex vivo*)). Right panel: Kaplan–Meier analysis of brain metastasis-free survival in control low BMI and reversed low BMI mice (n=6 each) [log-rank (Mantel–Cox test)]. **C.** *Ex vivo* quantification of metastasis in lung, liver and bone in control and reversed low BMI mice (n=4 each) by BLI at the endpoint (unpaired two-tailed t-test).

11. Supplementary Fig 5B-C – the authors show decrease in viability of the lung cancer cells upon treatment with really high concentrations of the Y5R antagonist (100μM). However, these cells are not stimulated with NPY. Do lung cancer cells secrete their own NPY?

Response: We appreciate the reviewer's comment. The aim of this experiment (now Supplementary Figure 6E-F) was to evaluate the dose-specific toxicity of Y5R antagonist in cancer cells under NPY-unstimulated conditions. Our data showed that higher doses of the Y5R

antagonist induced cell death compared to lower doses. Therefore, for potential clinical application, we selected a low dose for both *in vitro* and *in vivo* experiments. Additionally, we have included NPY ELISA data from cultured lung cancer cells, which show no detectable NPY secretion in the conditioned media, indicating the absence of autocrine signaling (see new data in revised Supplementary Figure 2 panel I; also see figure below).

Supplementary Figure 2: Low BMI-induced ghrelin activates neuronal NPY and promotes tumor growth. I. Quantification of NPY protein levels in CM derived from parental and brain-tropic lung cancer cell lines using ELISA ($n = 3$ individual experiment/group, unpaired two-tailed t-test).

12. Supplementary Fig 5D – in addition to lipid accumulation analysis, the authors should evaluate other features of the metastatic tissue from mice treated with Y5R antagonist, such as proliferation and cell death levels.

Response: We appreciate the reviewer’s valuable comment. In response, we have performed immunohistochemical staining for the Ki-67 marker to assess cancer cell proliferation in mouse brain metastatic tumor tissues from both the control and Y5R antagonist-treated groups. The results show that treatment with the Y5R antagonist significantly reduced cancer cell proliferation compared to the control group under low-BMI conditions. These findings further support the potential clinical application of MK-0557 in reducing lung cancer brain metastasis (see new data in revised Figure 5 panel J; also see figure below).

Figure 5. Low BMI reversibility or Y5R inhibition suppresses lung cancer brain metastasis. J. Left panel: Representative IHC staining of Ki-67 cells in metastatic brain tumor tissue derived from PBS- or MK-0557 ($n=5$ each) treated low BMI mice [Scale bar = 100 (Ki-67), insert (2x Zoom)]. Right panel: Dot plot showing quantification of Ki-67+ cells in metastatic brain tumor tissue using H-score (unpaired two-tailed t-test).

13. Method section – the description of the brain cell isolation and colony formation assay is too superficial to understand the procedure.

Response: We appreciate the reviewer's comment. In the revised manuscript, we have adequately addressed this concern.

General comments:

1. Overall, the language needs improvement. There are numerous grammatical errors (including the title) and many sentences are difficult to understand.

Response: As suggested, we have revised the manuscript and performed a comprehensive review to address grammatical errors, thereby improving the overall readability of the text.

2. Figures are not well organized and some of them span two pages. The description of the panels (e.g. left, bottom) often does not match their actual locations in the figures.

Response: We thank the reviewer for raising this point to our attention. In the revised version, we have adequately addressed this concern.

Reviewer #2

This manuscript investigates the molecular mechanisms under the low BMI and increased brain metastasis risk in lung cancer patients. The data show that low BMI increased ghrelin-induced secretion of neuronal NPY, which promotes metabolic reprogramming of tumors via Y5 receptors, making them more prone to fatty acid synthesis. Overall, this is an interesting study and the experimental approach in sound.

We thank the reviewer for highlighting the novelty and intriguing nature of our findings.

Major comments:

1. The canonical action of ghrelin on NPY neurons is based on p53/SIRT1 and AMPK/mTOR signaling pathways, at least at hypothalamic level. However, these molecular mechanisms, which have also a major (direct) impact on fatty acid metabolism (synthesis and oxidation; see below) have been totally ignored in this study, why? This should be explored, as well as the key transcriptional machinery modulating these effects, such as BSX, FoXO1 and CREB.

Response: We thank the reviewer for this comment. The primary aim of our study was to explore the impact of neuronal NPY on tumor cells expressing its receptor, NPY5R, to uncover underlying mechanisms and potential therapeutic targets. However, to address the reviewer's comment, we treated human primary neurons with or without ghrelin, in the presence or absence of a Sirt1 inhibitor (Ex527), and assessed signaling pathways, including NPY, via western blot. Our results showed that ghrelin treatment led to increased protein levels of NPY, SIRT1, pAMPK, and key transcription factors such as BSX, FoxO1, and CREB, consistent with previous studies (Lage *et al.*, 2010; Velasquez *et al.*, 2011). In contrast, treatment with the SIRT1 inhibitor alone, or in combination with ghrelin, resulted in decreased levels of these proteins compared to controls. Notably, NPY protein levels remained elevated under the combinatorial treatment (see figures below). These findings suggest that, in addition to the p53/SIRT1 and AMPK/mTOR pathways, other signaling pathways (PKC/ERK) activated by ghrelin may play a role in the induction and maintenance of neuronal NPY expression and secretion, as previously reported (Sieburth *et al.*, 2007; Kohno *et al.*, 2008; Cavalier *et al.*, 2015).

A. Human primary neuron were treated with or without recombinant ghrelin (0.25nM) in the presence or absence of SIRT1 inhibitor (Ex527; 1μM) and examined for the levels of indicated proteins, including NPY, by western blot ($n=3$ individual experiments, unpaired two-tailed t-test). **B.** Quantitative analysis of band intensities using ImageJ (unpaired two-tailed t-test).

2. In keeping with the former point, the increased ghrelin tone in neurons is known to be associated with decreased acetyl-CoA carboxylase activity and decreased Srebp1 and Fasn expression, as well as FAS activity. The overall effect is a malonyl-CoA reduction, therefore leading to increased CPT1 activity and fatty acid oxidation. This is opposite to the observed effects, with increased Fasn expression and fatty acid accretion. I agree these are tumor cells, but still molecular details are demanding Therefore, this should be investigated in deep and explained.

Response: We appreciate reviewer's suggestion. To address this question, we treated both primary neurons and cancer cells with or without recombinant ghrelin and assessed total SREBP2 and FASN at both mRNA and protein levels using qRT-PCR and western blot. Our results show that ghrelin treatment in neurons did not reduce total SREBP2 levels at either the mRNA or protein levels. However, FASN levels were reduced, correlating with decreased activated SREBP2 at both mRNA and protein levels. Strikingly, cancer cells did not respond to ghrelin treatment, as they express low to negligible levels of the ghrelin receptor (GHSR), as previously shown in Figure 2, panel C. Consequently, neither SREBP2 nor FASN levels changed in these cells at the mRNA or protein levels (see figures below). These results highlight the complexity of ghrelin signaling, which differs between neuronal and cancer cell types, likely due to variations in ghrelin receptor expression.

A-B. Human primary neuron and lung cancer cells were treated with or without recombinant ghrelin (0.25nM) and examined for the expression of total SREBP2 and FASN expression by qRT-PCR. β -Actin was used as a normalization control ($n=3$ individual experiments). **C-D.** Cells in panel A-B were examined for total SREBP2 and FASN expression by western blot. GAPDH was used as a normalization control ($n=3$ individual experiments). Quantitative analysis of band intensities using ImageJ (unpaired two-tailed t-test).

3. Y5 expression is increased upon NPY treatment. The molecular mechanism of this action needs to be investigated.

Response: We thank the reviewer for this valuable comment. To investigate the molecular mechanism underlying NPY-mediated Y5R upregulation, we explored potential regulatory factors for the Y5R gene. Our analysis identified a putative SREBP2 transcription factor binding site within the 5' region of NPY5R (-515 relative to TSS). Based on this finding, we hypothesized that SREBP2 plays a crucial role in mediating this effect. Furthermore, our clinical data from lung cancer brain metastatic patients revealed a significant positive correlation between the expression of Y5R and ERK5 as well as between ERK5 and SREBP2 (see in response to Reviewer's 3, Major comment #3; see also Figure 4, panel F). Therefore, to validate these findings, we ectopically expressed either ERK5 or SREBP2 in lung cancer cells (CMT167), with or without recombinant NPY, in the presence or absence of ERK5 (ERK5-IN-1) or SREBP (Fatostatin HBr) inhibitors. We found that overexpression of ERK5 or SREBP2 significantly upregulated Y5R receptor expression compared to controls and this upregulation was markedly reduced upon treatment with their respective inhibitors (see new data in revised Figure 4, panel I; also see figure below). Collectively, these results suggests that under low-BMI condition, NPY aberrantly activates and regulates Y5R receptor expression via the ERK5/SREBP2 axis in an autocrine manner. This signaling cascade induces FAO signaling in cancer cells, promoting lipogenesis and enhancing brain metastatic tumor growth.

Figure 4. Low BMI-induced neuronal NPY upregulates lipogenesis via Y5R/SREBP2/FASN signaling axis to fuel tumor cell growth. I. LL2 lung cancer cells were transfected with or without Erk5 or Srebp2 expression plasmids and treated with or without Srebp inhibitor (Fatostatin HBr) or Erk5 inhibitor (ERK5-IN-1), in presence or absence of recombinant Npy and examined for activated Srebp2, Npy5r by western blot. GAPDH was used as a normalization control ($n=3$ individual experiments, unpaired two-tailed t-test). Lower, right panel: Quantitative analysis of band intensities using ImageJ (unpaired two-tailed t-test).

4. Line 94, I think it should read Fig. 1E instead of Fig. 1D.

Response: In the revised version, we have corrected the typo error.

Reviewer #3

The inverse relationship between obesity and NSCLC outcome termed obesity paradox has been confounded by many factors including anti-diabetic drug metformin use in obese patients and BMI not being able to distinguish abdominal obesity. This is an active area of investigation. Tyagi et al performed a detailed characterization of low BMI-associated lung-to-brain metastasis and attributed it to increase in neuronal neuropeptide Y (NPY) secretion via ghrelin-GHSR receptor activation reprogramming the cancer cell energy metabolism. Increase in serum ghrelin in low BMI condition activates NPY which supports Y5R(high) tumor cells and prime the cells for brain metastasis. The manuscript is well written and underlying biological findings are novel.

We thank the reviewer for their recognition of the novelty of the observations included in the original report.

Major comments:

1. Figure 1A: presents the data as % incidence, but there is no apparent consideration for time in the analysis. If incidence is the endpoint, then a Kaplan Meir analysis would be more appropriate.

Response: We appreciate the reviewer's insightful comment. While we agree that a Kaplan-Meier analysis would offer a more comprehensive representation, we regret that survival (time-to-event) data were not available from the retrospective studies for these groups. However, in the initial submission, we included brain metastasis-free survival data for clinical samples of lung cancer brain metastatic patients based on BMI (see Supplementary Figure 1, panel A). We hope this clarification adequately addresses the concern.

2. For all the in vivo analysis of brain metastasis, H&E images should be provided to corroborate key BLI based conclusions (e.g. Fig 1E, 3I, 5G).

Response: We thank the reviewer for this comment. To address this important question, we have performed H&E staining on paraffinized brain sections of all the representative images (Scale bar: 50 μ M) corresponding to Fig 1E, 3I (now in Supplementary Figure 4G) and 5G (now in Supplementary Figure 7); also see figures below.

Figure 1. Low body mass index (BMI) is associated with increased brain metastasis in lung cancer.
E. Lower panel: Representative images of hematoxylin and eosin (H&E) on brain metastatic tumor lesions derived from normal and low BMI mice (Scale bar = 50 μ m).

Supplementary Figure 4: Neuronal NPY enhances brain metastatic growth via Y5R receptor signaling. **G.** Representative images of hematoxylin and eosin (H&E) on brain metastatic tumor lesions derived from Scr-KO and Npy5r-KO normal and low BMI mice (Scale bar = 50 μ m).

Supplementary Figure 7: Low BMI reversibility and Y5R inhibition mitigate brain metastasis. **A.** Representative images of hematoxylin and eosin (H&E) on brain metastatic tumor lesions derived from low BMI and low BMI reversed mice from Figure 5, panel A (Scale bar = 50 μ m). **B.** Representative images of hematoxylin and eosin (H&E) on brain metastatic tumor lesions derived from PBS and MK-0557-treated low BMI mice from Figure 5, panel G (Scale bar = 50 μ m).

- The manuscript relies heavily on GSE200563, which includes pairs of primary lung cancers and brain metastasis. However, in this prior study/dataset, the authors include separate gene expression data from the tumor stroma microenvironment as well as the tumor core. It is not clear how the present authors analyzed these data to reach their conclusions (e.g. in Fig. 3G).

Response: We appreciate the reviewer’s comment. In the initial analysis, we analyzed both tumor core and stroma expression data from metastatic samples using the GSE200563 dataset. However, the stroma data was less informative due to its mixed cell-type composition. Therefore, we prioritized brain metastasis tumor core data for gene correlation analysis to derive our conclusions.

Furthermore, in the initial submission of Figure 3G, we inadvertently used primary tumor core data for the correlation plots. This error has been corrected by using the brain metastatic (LB) tumor core data and re-plotting the correlation plots (see new data in revised Figure 3, panel E). We apologize for this oversight.

Our revised data demonstrate a significant correlation between ERK5 (MAPK7) and NPY5R expression, in contrast to MAPK8 (JNK) or MAPK14 (p38) or other MAPK family members (ERK1/2/3), in brain metastatic samples (see figure below). These findings are further validated by western blots using protein lysates from recombinant NPY- and vehicle-treated cancer cells [see response to *Minor comment #8*]. Overall, these results identify ERK5 as a key driver of the NPY/Y5R signaling axis during the initial stages of metastatic colonization in the brain.

Figure 3. Low BMI-mediated neuronal NPY promotes brain metastasis via Y5R receptor signaling.
F. Pearson correlation analysis between Y5R receptor expression and MAPK family members isoforms (ERK5, P38, JNK) in lung cancer brain metastatic patients ($n=27$) using GEO dataset (GSE200563).

- Line 575 authors mention “freshly isolated brain cells (human and mouse)” for conditioned medium in neuron culture. It is however not clear whether human/mouse neurons were used in experiments shown in figure 2B, 2C and 2D. How were fresh human brain cells isolated and cultured? What age were mouse neurons isolated? If mouse neurons were isolated from prenatal mice or day 0 pups, were the expression level for ghrelin-GHSR receptor and NPY like those of

adult mouse neurons? If adult mice were used for neuron isolation, what was the BMI status for these mice?

Response: We apologize for missing the description of the source of primary neurons in the initial submission. For this study, we utilized commercially available iPSC-derived human primary neurons from BrainXell to generate all relevant data and their associated conditioned media, which were cultured in their respective media with or without the indicated recombinant protein treatment. This information has now been corrected in the Methods section of the revised manuscript.

5. The impact of conditioned medium is shown in BrM cell lines in figure 2C. BrM lines are shown to have low GHSR expression in figure 2B. Since these cell lines have already been through the brain microenvironment once, it would be important to know the levels in parental lines.

Response: We agree with the reviewer's comment and have included GHSR expression data for both parental human lung cancer cell lines (H2030 and PC9). As expected, both parental cell lines showed low to negligible GHSR expression compared to human primary neurons, consistent with the expression levels observed in their brain-tropic variants (see new data in revised Supplementary Figure 2, panel B; also see figure below) in the revised manuscript.

Supplementary Figure 2: Low BMI-induced ghrelin activates neuronal NPY and promotes tumor growth. B. Relative expression of ghrelin receptor (GHSR) in parental lung cancer cell lines (H2030, PC9) compared to human primary neurons (used as a control) by qRT-PCR. β -Actin was used as a normalization control ($n = 3$ individual experiment, unpaired two-tailed t-test).

6. Line 423: The authors mention "Glycolysis is less important for GBM metabolism". However, GBM tumors have significant up-regulation of both glucose and fatty acid oxidation pathways (<https://pubmed.ncbi.nlm.nih.gov/33854970/>) and this depends on aggressiveness in GBM. Please restructure the sentence to reflect that.

Response: In the revised version, we have incorporated reviewer's suggestion.

There is also lack of careful consideration of causality and bias on metformin use. There is no discussion of metformin use in obese patients. Metformin directly inhibits ghrelin production and causes weight loss in both diabetic and non-diabetic individuals. What is the result of impact of inherent less ghrelin in these individuals, how can this be separated from additional impact of low BMI, and what portion of the phenotype is likely cell autonomous due to low expression of this receptor in high BMI individuals?

Response: We appreciate the reviewer's suggestion. Emerging evidence suggests that the effects of metformin on ghrelin levels remain controversial. One study report that metformin increases plasma ghrelin levels by 24% in individuals with Type 2 diabetes (T2DM) (Doogue *et al.*, 2009), while other studies indicate that metformin either reduces plasma ghrelin levels or shows no significant association with ghrelin levels in T2DM patients (English *et al.*, 2007; Ida *et al.*, 2017). Furthermore, individuals with T2DM generally exhibit lower ghrelin levels compared to non-T2DM controls (Poykko *et al.*, 2003; Chen *et al.*, 2017). However, the impact of inherently lower ghrelin levels and the use of metformin in these individuals, particularly in the context of brain metastasis, is not yet fully understood. Notably, recent data indicate that the incidence of brain metastases was significantly lower in diabetic melanoma patients exposed to metformin (Augustin *et al.*, 2023; LeCompte *et al.*, 2018). Further investigation is warranted to better understand the underlying mechanism(s), which is beyond the scope of the current study. We hope this clarification addresses the reviewer's concern, which has been included in the discussion section (Page#14, Line #12) of the revised manuscript.

Minor comments:

1. Line 159: Replace "we quired" with "we queried".

Response: In the revised version, we have corrected the typo error.

2. Line 515: Replace "then imaged were evaluated by" with "then images were evaluated by"

Response: In the revised version, we have corrected the typo error.

3. Figure 1E: Do the ROIs used for all in-vivo analysis placed to reflect similar cranial area for all animals? Do the ROIs used for all ex-vivo analysis have the same dimensions?

Response: We appreciate the reviewer's comment. To ensure consistency in our *in vivo* and *ex vivo* analysis, we reapplied the default region of interest (ROI) dimensions to all animals including representative images. While we observed minor differences in cranial area due to uneven animal posture, it didn't not affect the overall ROI measurements. Importantly, the conclusions of our findings remained unchanged after reanalyzing the data using these consistent ROI dimensions.

4. Figure 3A: P values are not defined.

Response: In the revised version, we have incorporated reviewer's suggestion.

5. Figure 2C: Increased proliferation in neuron CM is only shown for H2030BrM. Was the same phenotype observed for other cell lines? Also, what is the time frame for the clonogenic experiments (endpoint and treatment interval). How is it possible to see an effect, if the treatments are presumably transient?

Response: We appreciate the reviewer's comment. Increased proliferation in CM from neurons was also observed for other cell lines (PC9BrM) as shown below. Furthermore, we apologize for the previous unclear explanation of the clonogenic experiments. Briefly, cells were treated with or without conditioned media (CM) derived from indicated brain cell types, or recombinant NPY, or ghrelin to control media and allowed to grow for 7 days. The media were replenished every 3rd day. On day 7, the colonies were fixed with ethanol for 30 min, stained with Crystal Violet (CV) for 20 min., and then the number of colonies was counted manually.

A. PC9BrM lung cancer cells were treated with or without CM derived from human primary neurons and examined for and growth by colony formation assay ($n = 3$ individual experiment, unpaired two-tailed t-test).

6. Figure 2D x-axis labeling is missing.

Response: In the revised version, we have included the X-axis labeling.

7. Figure 4D: Western blot for total SREBP2 is missing.

Response: As per reviewer's recommendation, we have included the western blot data for total SREBP2 in Figure 4D, using the remaining protein lysate derived from H2030BrM cells treated with or without recombinant NPY protein. Additionally, we have provided the corresponding quantitative data as a graph, along with statistical analysis, in the revised version (see revised Figure 4, panel D; also see figure below).

Figure 4. Low BMI-induced neuronal NPY upregulates lipogenesis via Y5R/SREBP2/FASN signaling axis to fuel tumor cell growth. D. Upper panel: Assessment of lipogenic regulator, SREBP2 along with Y5R^{high}-specific key oncogenic signaling in control (Scr-KO) and y5r-KO CMT167 lung cancer cells treated with or without recombinant NPY protein (100 nM) by western blot. GAPDH was used as a control for normalization ($n=3$ individual experiments). Lower panel: quantitative analysis of band intensities using ImageJ (unpaired two-tailed t-test).

8. Figure 4E: There is difference in intensity within the western blot band for pSREBP2. Did the authors corroborate the findings in a replicate?

Response: We appreciate the reviewer's comment. In the revised version, we repeated the entire western blot using protein lysates from recombinant NPY- and vehicle-treated cancer cells to rule out any discrepancy and also provided the quantitative data as a graph with statistical analysis (see new data in revised Figure 4, panel E; also see figure below).

Figure 4. Low BMI-induced neuronal NPY upregulates lipogenesis via Y5R/SREBP2/FASN signaling axis to fuel tumor cell growth. E. Upper panel: H2030BrM cancer cells were treated with or without recombinant NPY protein (100nM) and examined for activated ERK5, SREBP2 by western blot. GAPDH was used as a control for normalization ($n=3$ individual experiments). Lower panel: quantitative analysis of band intensities using ImageJ (unpaired two-tailed t-test).

9. Figure 4G: What is the explanation for reduced pErk in the y5rKO+rNPY treated relative to y5rKO unstimulated? A few more time points for this analysis might reveal the proper kinetics of this activation.

Response: We appreciate the reviewer's comment. As addressed in response to *Major Comment #3*, in the initial submission of Figure 3G, we inadvertently used primary tumor core data for the correlation plots. This error has been corrected by using the brain metastatic (LB) tumor core data and re-plotting the correlation plots. We sincerely apologize for this oversight. Our revised data show a significant correlation between ERK5 (MAPK7) and NPY5R expression, in contrast to MAPK8 (JNK) or MAPK14 (p38) or other MAPK family members (ERK1/2/3), in brain metastatic samples (see response to *Major comment #3*).

Because of this correction, we repeated the entire western blot experiments using recombinant NPY and ERK5 inhibitor-treated ScrKO and Y5rKO cancer cells (see response to *Minor comment #8*; also see *Reviewer's 1, Minor comment #9*). Additionally, we validated this further by ectopically expressing ERK5 or SREBP2 in cancer cells with or without recombinant

NPY, in the presence or absence of ERK5 (ERK5-IN-1) or SREBP (Fatostatin HBr) inhibitors (see response to *Reviewer's 2, Major Comment #3*).

Following the reviewer's suggestion, we performed time kinetics of ERK5 activation in Y5rKO cells with or without recombinant NPY treatment. Our results indicate no change in total ERK5 levels, but a reduction in activated Erk5 levels at 12 hours post-treatment. While the reason for the reduced basal expression of ERK5 remains unclear, there could be couple of possible explanations:

1. The loss of a receptor (Y5R) might induce cellular stress responses (e.g. activation of stress kinases or unfolded protein response pathways), which could impact protein stability or synthesis, leading to a decrease in ERK5 levels.
2. The loss of a Y5R receptor might trigger negative feedback loops or the activation of alternate pathways that downregulate ERK5 expression.
3. Knockout of Y5R may affect post-translational modifications (e.g., phosphorylation, ubiquitination) of ERK5 that could influence its stability or activity over time.

Further research is warranted to explore these possibilities and fully elucidate the underlying mechanisms.

A. Left panel: Y5r-KO cancer cells were treated with or without recombinant NPY (100nM) and examined for total ERK5 and activated ERK5 in time-dependent manner by western blot ($n=3$ individual experiments, unpaired two-tailed t-test). GAPDH was used as a control for normalization ($n=3$ individual experiments). Right panel: quantitative analysis of band intensities using ImageJ (unpaired two-tailed t-test).

Point-by-point response to Reviewer's Comment's

We thank all the Reviewers for going through our revised manuscript which have enabled us to greatly improve the manuscript. As described below, we have now addressed all the critiques and suggestions from the reviewers and hope that the manuscript can be now considered for publication in *Nature Communications*. The responses appear in dark-red font. Changes in the manuscript have been highlighted yellow to facilitate review.

Reviewer #1

1. Figure 2L – interpretation of the data presented in this panel is not correct. Typically, accumulation of cells in G1 phase indicates inhibition of cell proliferation rather than its stimulation. The results of the cell cycle analysis may depend on the time after stimulation. Perhaps the assay that measures cumulative DNA synthesis, such as BrdU or EdU incorporation, will give more reliable results.

Response: We appreciate the reviewer's concern. As per the reviewer's suggestion, we have repeated the cell cycle analysis using a BrdU kit (BD Pharmingen™ FITC BrdU Flow Kit). Our new results showed that NPY treatment (24 hours) significantly increased the S phase and decreases the G₀/G₁ phase of the cell cycle compared to controls, contradicting the initial cell cycle data submitted during the first revision. Therefore, to address this discrepancy, we have also repeated the PI-based cell cycle assay using a commercially available kit (Cell Cycle Phase Determination Kit, Cayman Chemical), and we found the consistent results with those obtained from the BrdU assay. We anticipate that the previous inconsistency stemmed from the shorter treatment duration post-stimulation and the manual execution of the PI-based analysis, which could have introduced variability in data interpretation. Overall, the new results further support the role of the NPY/NPY5R signaling axis in promoting cancer cell proliferation (see the updated data in revised Figure 2, panel L, as well as the figure below).

Figure 2: Low BMI promote tumor growth via ghrelin activated neuronal NPY. L. Left panel: Representative dot plot of measurement of BrdU uptake by flow cytometry in H2030BrM cells treated with or without recombinant NPY (100nM). Right panel: Quantification of G0/G1, S, and G2/M cell cycle fractions ($n = 3$ individual experiment, unpaired two-tailed t-test).

- Supplementary figure 4F is missing Scr KO and Y5R KO cell lines without NPY treatment. It would be interesting to know if Y5R KO affects basal cell proliferation.

Response: We apologize for missing the Scr-KO/Y5r-KO data without NPY treatment. To substantiate our findings, we have now repeated the experiments presented in Supplementary Figure 4F, using WT, Scr-KO, and Y5r-KO cancer cells, both with and without recombinant NPY treatment. Our results indicate that while Y5r-KO cancer cells without NPY treatment exhibited slight decrease in basal cell proliferation and growth, it does not show any significant differences compared to its NPY-treated counterparts (see the updated data in revised Figure 4, panel F, as well as the figure below).

Supplementary Figure 4: Neuronal NPY enhances brain metastatic growth via Y5R receptor signaling. F. Cells from panel D were treated with or without recombinant NPY protein (100nM) and examined for cell proliferation by MTS assay (left panel) and growth by colony formation assay (right panel) ($n=3$ individual experiments, unpaired two-tailed t-test).

- The title requires revision. There is no such thing as “low BMI lung cancer”.

Response: We thank the reviewer for pointing this out. To clarify, we have revised the title to “Ghrelin activates neuron to upregulate NPY which promotes brain metastasis in lung cancer patients with low BMI”.

- Upon addition of new data, the supplementary figures are not referenced in the text in order. Supplementary figure 2A and Supplementary figure 3 should be combined and presented as Supplementary figure 2, while current Supplementary Figure 2 B-K should become Supplementary figure 3.

Response: We appreciate the reviewer's helpful suggestion. To address this concern, we have rearranged the supplementary figures as recommended in the revised manuscript and have revised the figures labeling accordingly.

5. Figure 2J-K – the labeling is confusing, as only panel J is labeled “H2030BrM”, while both panels J and K present data obtained using this cell line. Moreover, the difference between the proliferation assays shown in panels J and K is not stated in the legend (short term vs long term treatment).

Response: We appreciate the reviewer's observation. To clarify, we have revised the labeling in the figure and the figure legend in the revised manuscript to ensure the distinction between short term vs long term NPY treatment.

6. Figure 2M – it is not mentioned which ERK is being detected. Similarly, in the text describing Supplementary figure 2K it should be mentioned that the inhibitor used in the experiment targets ERK1/2. Lastly, NPY at the molecular weight of 11kDa should be labeled as pro-NPY, since the molecular weight of mature NPY is lower.

Response: We thank the reviewer for pointing this out. To address these concerns, we have clearly specified that ERK1/2 is being detected. Additionally, in the description of Supplementary Figure 2K, we have mentioned that the inhibitor used targets ERK1/2. Furthermore, we have revised the labeling of NPY to indicate it as pro-NPY in the revised manuscript.

7. Figure 3D – GSEA analysis shows enrichment of genes involved in “neuropeptide receptor activity”. However, the authors refer to it specifically as enrichment in NPY receptor activity. As there are many neuropeptides, it is not clear which interpretation is correct.

Response: We thank the reviewer for this valuable observation. To clarify, we have revised the text to emphasize that the enrichment pertains to neuropeptide receptor activity in general, rather than exclusively to NPY receptor activity. This revision ensures a more accurate interpretation of the results.

8. Lines 178-181 – the sentence describing Supplementary figure 4A is not clear.

Response: We apologize for the lack of clarity. To enhance clarity, we have revised and corrected the figure description in the revised manuscript as “Additionally, increased expression of Y5R is associated with poor overall survival in lung cancer patients, attributed to its elevated levels in metastatic lung tumors compared to normal and primary lung tumors. In contrast, in breast cancer, Y5R expression is correlated with better overall survival, with higher expression observed in normal and primary tumors compared to metastatic breast tumors (**Supplementary Fig. 4A**)”.

9. Lines 184-186 – the sentence describing Supplementary figure 4B is not clear.

Response: We apologize for the lack of clarity and have revised the figure description in the revised manuscript as “To further validate this observation, we analyzed multiple GEO cohort datasets, including clinical and preclinical samples of lung, breast, and melanoma metastatic tumors, as well as primary tumor xenografts (GSE123904 /GSE48433 /GSE14020/GSE50493). We found that Y5R was selectively expressed in metastatic tumors, primarily in the brain, in lung cancer, in contrast to metastatic tumors from breast and melanoma cancers (**Supplementary Fig. 4B**)”.

10. Line 221 and Figure 3 panel I – Yr5 – typo.

Response: The typo error has been corrected in the revised manuscript.

11. Line 264 – the sentence is not clear. Please add “... in NPY-treated parental lung cancer cells”.

Response: Thank you for the suggestion. We have included the statement “... in NPY-treated parental lung cancer cells” for greater clarity in the revised manuscript.

12. Legend for Figure 5 – the descriptions of panels D and E are switched.

Response: We apologize for the oversight. The descriptions of panels D and E have been corrected in the revised manuscript to ensure clarity and accuracy.

13. While describing the lack of the effect of MK-0557 on the mice body weight, the authors do not mention that the treatment was done in mice with already low body weight. Y5R inhibition may still affect weight of normal body weight mice. This should be discussed.

Response: We thank the reviewer for this valuable comment. In the revised manuscript, we have added a sentence in the discussion section (see page 16; lines 497-502) emphasizing that the effects of Y5R inhibition on body weight may vary in normal-weight mice, and further investigation is warranted to assess this in mice with normal body weight.

14. In the discussion section, the authors provide the full name of many terms abbreviated in earlier parts of the manuscript, such as NPY. This is not necessary.

Response: We thank the reviewer for pointing this out. In response, we have revised the manuscript to avoid repeating the full names of terms that were previously abbreviated.

15. Throughout the manuscript the authors use the phrase “Y5R receptor”. In the Y5R abbreviation, R stands for “receptor”. Thus “Y5R” should be sufficient.

Response: We appreciate the reviewer for bringing this to our attention. In response, we have revised the manuscript to eliminate the redundant use of the term 'receptor' when referring to 'Y5R.' The updated manuscript now consistently uses 'Y5R'.

Reviewer #2

1. All my comments have been addressed. This is an interesting and technically sound manuscript.

Response: We sincerely thank the Reviewer for acknowledging the value of the study.

Reviewer #3

1. The authors addressed the question of neuron conditioned medium by providing the source of neurons and included it in the revised Methods section.

Response: We thank the reviewer for acknowledging our responses.

2. For the comment on GHSR expression data for parental cell lines, authors showed lower expression in parental as compared to neurons. But is it lower in BrM lines than the parental lines,

which was the original comment. To address this, could the authors show BrM and parental lines on the same graph?

Response: We thank the reviewer for raising this important point. To provide a clearer comparison, we have now repeated the experiments, including both parental and brain-tropic lung cancer cell lines within the same graph for GHSR expression levels. Our results confirm that both parental and brain-tropic lung cancer cell lines exhibit low GHSR expression levels compared to neurons, as shown in Figure 2C and Supplementary Figure 2B, aligning with our original conclusions. Please find the updated figure below.

A. Normalized expression of ghrelin receptor (GHSR) in parental (PC9, H2030) and brain-tropic (PC9-BrM, H2030-BrM) lung cancer cell lines by qRT-PCR. β -Actin was used as a normalization control ($n = 3$ individual experiment, unpaired two-tailed t-test).

- Line 423 in original manuscript has been satisfactorily modified based on current literature for GBM metabolism.

Response: We appreciate the reviewer's acknowledgment and are pleased that our responses addressed their concerns.

- For metformin use, the authors have included relevant publications in the discussion. However, since metformin has anti-tumor effects in lung cancer (<https://doi.org/10.1016/j.jtho.2020.08.021>, <https://doi.org/10.18632/oncotarget.17066>), it would be important to include it as a confounding factor in context of cancer. Please include a sentence explaining role of metformin in lung cancer in the discussion.

Response: We thank the reviewer for this valuable suggestion. In the revised manuscript, we have included a sentence in the discussion section (see page 14, lines 435-438) to acknowledge the potential role of metformin as a confounding factor in lung cancer.

- All additional comments have been addressed.

Response: We sincerely appreciate the reviewers' positive feedback on our revised manuscript.